# TAMP-S2GCNets: Coupling Time-Aware Multipersistence Knowledge Representation with Spatio-Supra Graph Convolutional Networks for Time-Series Forecasting

**Yuzhou Chen**[1,3]  **Ignacio Segovia-Dominguez**[2,4]  **Baris Coskunuzer**[2]  **Yulia R. Gel**[2,5]

[1]Department of Electrical and Computer Engineering, Princeton University
[2]Department of Mathematical Sciences, University of Texas at Dallas
[3]Lawrence Berkeley National Laboratory
[4]Jet Propulsion Laboratory, Caltech
[5]National Science Foundation
`yc0774@princeton.edu`
`{Ignacio.SegoviaDominguez, coskunuz, ygl}@utdallas.edu`

## Abstract

Graph Neural Networks (GNNs) are proven to be a powerful machinery for learning complex dependencies in multivariate spatio-temporal processes. However, most existing GNNs have inherently static architectures, and as a result, do not explicitly account for time dependencies of the encoded knowledge and are limited in their ability to simultaneously infer latent time-conditioned relations among entities. We postulate that such hidden time-conditioned properties may be captured by the tools of multipersistence, i.e., a emerging machinery in topological data analysis which allows us to quantify dynamics of the data shape along multiple geometric dimensions. We make the first step toward integrating the two rising research directions, that is, time-aware deep learning and multipersistence, and propose a new model, Time-Aware Multipersistence Spatio-Supra Graph Convolutional Network (TAMP-S2GCNets). We summarize inherent time-conditioned topological properties of the data as time-aware multipersistence Euler-Poincaré surface and prove its stability. We then construct a supragraph convolution module which simultaneously accounts for the extracted intra- and inter-dependencies in the data. Our extensive experiments on highway traffic flow, Ethereum token prices, and COVID-19 hospitalizations demonstrate that TAMP-S2GCNets outperforms the state-of-the-art tools in multivariate time series forecasting tasks.

## 1 Introduction

Multivariate time series forecasting plays an integral role in virtually every aspect of societal functioning, from biosurveillance to financial analytics to intelligent transportation solutions. In the last few years, Graph Convolutional Networks (GCNs) have emerged as a powerful alternative to more conventional time series predictive models. Despite their proven success, GCNs tend to be limited in their ability to simultaneously infer latent temporal relations among entities (such as correlations both within a time series and in-between time series or the joint spatio-temporal dependencies). More generally, most existing GCNs architectures are inherently static and as such, do not explicitly account for time-conditioned properties of the encoded knowledge about the complex dynamic phenomena.

At the same time, recent studies on integrating shape properties of the complex data into deep learning (DL) models indicate that topological representations, obtained using the tools of single parameter persistence, can bring an invaluable insight into the system organization and enhance the resulting graph learning mechanisms (Hofer et al., 2019; Carrière et al., 2020; Carlsson & Gabrielsson, 2020; Horn et al., 2021). (By shape here we broadly understand data properties which are invariant under continuous deformations, e.g., stretching, bending, and twisting). However, in many appli-

cations, particularly, involving spatio-temporal processes, the data exhibit richer structures which cannot be well encoded with a single parameter persistence. We postulate that many critical hidden time-conditioned interrelations which are inaccessible with other methods can be captured by the emerging machinery of *multipersistence*. Multipersistence, or multiparameter persistence (MP) generalizes the notion of single parameter persistence to a case when dynamics of the inherent data shape is discerned along multiple geometric dimensions (Carlsson & Zomorodian, 2009). Despite its premise, applications of MP in any discipline remain nascent at best (Riess & Hansen, 2020; Kerber, 2020).

We make the first step on a path of bridging the two emerging directions, namely, time-aware DL with time-conditioned MP representations of complex dynamic phenomena. By time-conditioned MP representations, we mean the most salient topological properties of the data that manifest themselves over time. To summarize such time-conditioned topological properties, we first introduce a dynamic Euler-Poincaré surface as a new MP summary. We then propose a directed multilayer supra graph abstraction to represent a sequence of time-varying objects and develop a supragraph convolution module which allows us to simultaneously learn co-evolving intra- and inter-dependencies (i.e., spatial and temporal correlations) in the complex high-dimensional data.

The key novelty of this paper are summarized as follows:

- This is the first work to bridge the concepts of MP with the time-aware learning paradigm. Applications of MP in any field of study are currently nascent.
- We introduce a new time-aware multipersistence invariant, a *dynamic Euler-Poincaré surface*. We prove its stability and show its substantial computational gains and high utility for encoding the time-conditioned knowledge.
- We propose a mathematical abstraction of directed multilayer supra graph for time-conditioned knowledge representation and construct a new Time-Aware Multipersistence Spatio-Supra Graph Convolutional Network (TAMP-S2GCNets) which simultaneously learns latent temporal inter- and intra-relations among entities in the complex high-dimensional data.
- We perform expansive forecasting experiments, in application to highway traffic flow, Ethereum token prices, and COVID-19 hospitalizations. Our findings demonstrate superior predictive performance, versatility and computational efficiency of TAMP-S2GCNets, compared to the state-of-the-art methods in multivariate time series forecasting.

## 2 RELATED WORKS

**Multipersistence** Despite that MP demonstrates very promising results in terms of improving accuracy, tractability and robustness, applications of MP in ML are virtually non-existent (Wright & Zheng, 2020; Riess & Hansen, 2020; Kerber, 2020). Some notable efforts in the direction to develop MP summaries which are suitable for integration with ML models include Multiparameter Persistence Kernel of Corbet et al. (2019), Multiparameter Persistence Landscapes (MP-L) of Vipond (2020), and Multiparameter Persistence Images (MP-I) of Carrière & Blumberg (2020) which are based on the concept of slicing, that is, restricting the MP module to an affine line (or single parameter persistence) (Cerri et al., 2013; Landi, 2014). Such slicing methods enjoy a number of important stability guarantees but tend to be computationally expensive even in static scenarios, which makes them infeasible for time series forecasting tasks. In turn, the most recent results of Beltramo et al. (2021) (i.e., Euler Characteristic Surfaces) and Coskunuzer et al. (2021) (i.e., Multiparameter Persistence Grids) introduce pointwise MP representations, in application to static point clouds and graphs. Such pointwise representations are weaker invariants but are substantially more computationally efficient. Integration of pointwise representation with ML models has not been yet investigated. Here we propose the first time-aware pointwise MP representation, *a dynamic Euler-Poincaré surface*, derive its theoretical properties and integrate it with GCN in time series forecasting tasks.

**Spatial-Temporal Graph Models and Forecasting** Recent studies (Li et al., 2018; Yu et al., 2018; Yao et al., 2018) introduce graph convolution methods into spatio-temporal networks for multivariate time series forecasting which, as a result, allows for better modeling of dependencies among entities (Wu et al., 2019; Bai et al., 2020; Cao et al., 2020) and handling data heterogeneity. Despite the GCN successes, designs of the existing spatial-temporal GCNs largely rely on the pre-defined graph structures. As such, GCNs are restricted in their ability to explicitly integrate time dimension into the knowledge representation and learning mechanisms, thereby, limiting model adaptivity

to the dynamic environments and requiring more frequent retraining. Most recently, Chen et al. (2021) propose a time-aware GCN, Z-GCNETs, for time series forecasting which integrates zigzag persistence images based on a single filtration, as the primary time-conditioned topological representation. In general, the zigzag concept can be combined with MP, but it requires more fundamental advances in the theory of algebraic topology. As such, our time-aware MP learning approach may be viewed as complementary to zigzag persistence, while considering time-changing connections of graph structures in dynamic networks.

## 3 TIME-AWARE MULTIPERSISTENCE EULER CHARACTERISTIC SURFACES

**Spatio-Temporal Graph Construction** We define a spatial network at time step $t$ as $\mathcal{G}_t = (\mathcal{V}_t, \mathcal{E}_t, A_t, \boldsymbol{X}_t)$, where $\mathcal{V}_t$ is a set of nodes and $\mathcal{E}_t$ is a set of edges. We let $|\mathcal{V}_t| = N$ and $|\mathcal{E}_t| = M_t$. The adjacency matrix $A_t \in \mathbb{R}^{N \times N}$, and $\boldsymbol{X}_t = \{x_{t,1}, x_{t,2}, \ldots, x_{t,N}\}^\top \in \mathbb{R}^{N \times F_N}$ is the node feature matrix with feature dimension $F_N$. To construct the spatial network $\mathcal{G}_t$, we can build the adjacency matrix $A_t$ based on (i) the prior knowledge of graph structure: first-order neighbours, i.e., $A_{t,uv} = 1$ if the node $u$ and node $v$ have a connection in the dynamic graph at the time step $t$; and (ii) the Radial Basis Function (RBF): degrees of similarity between instances (i.e., nodes) in $\boldsymbol{X}_t$, i.e., $A_{t,uv} = 1_{\exp(-||x_{t,u}-x_{t,v}||^2/\gamma) \leq \epsilon}$, where $\gamma$ denotes the length scale parameter and $\epsilon$ denotes the threshold parameter filters noisy edges. Let $T$ be the total number of time steps. Given a sequence of observations on a multivariate variable, $\boldsymbol{\mathcal{X}} = \{\boldsymbol{X}_1, \boldsymbol{X}_2, \ldots, \boldsymbol{X}_T\} \in \mathbb{R}^{N \times F_N \times T}$ with $T$ timestamps and $F_N$ node attributes, we construct spatio-temporal networks $\boldsymbol{\mathcal{G}} = \{\mathcal{G}_1, \mathcal{G}_2, \ldots, \mathcal{G}_T\}$ via either prior knowledge of network structure or applying RBF to the node feature matrix.

**Single Parameter Persistence** Persistent homology (PH) based on one parameter discerns shape of the complex data along a single geometric dimension. The goal is to select some suitable parameter of interest and then to study a graph $\mathcal{G}_t$ not as a single object, but as a sequence of nested subgraphs, or a *graph filtration* $\mathcal{G}_t^1 \subseteq \mathcal{G}_t^2 \ldots \subseteq \mathcal{G}_t^m = \mathcal{G}_t$, induced by this evolving scale parameter. Armed with such filtration, we can then assess which structural patterns (e.g., loops and cavities) appear/disappear and record their lifespans. To make the counting process more efficient and systematic, we build a simplicial complex $\mathbf{C}_t^i$ from each subgraph $\mathcal{G}_t^i$, resulting in a filtration $\mathbf{C}_t^1 \subseteq \mathbf{C}_t^2 \ldots \subseteq \mathbf{C}_t^m$ (e.g., clique complexes). For example, to construct such filtration, a common method is to consider a filtering function $f : \mathcal{V}_t \mapsto \mathbb{R}$ and the corresponding increasing set of thresholds $\{\alpha_i\}_1^m$ such that $\mathbf{C}_t^i = \{\Delta \in \mathbf{C}_t : \max_{v \in \Delta} f(v) \leq \alpha_i\}$. The resulting construction is called a *sublevel set filtration* of $f$, and $f$ can be selected, for instance, as degree, centrality, or eccentricity function (Hofer et al., 2020; Cai & Wang, 2020). Similarly, $f$ can be defined on the set of edges $\mathcal{E}_t$. More details on a single filtration PH can be found in Appendix B.

**Time-Aware Multiparameter Persistence (TAMP)** Data in many applications, particularly, involving spatio-temporal modeling, might be naturally indexed by multiple parameters, e.g., real time traffic flow and optimal route in urban transportation analytics. Alternatively, the primary focus might be on discerning shape properties of the complex data along multiple dimensions. For instance, to better predict cryptocurrency prices and manage cryptomarket investment performance, we may need to evaluate structural patterns in cryptoasset dynamics not along one dimension but simultaneously along the volume of transactions and transaction graph betweenness, as a measure of joint perception of the cryptomarket volatility among the key investors. Such multidimensional analysis of topological and geometric properties can be addressed using generalization of PH based on a single filtration to a *multifiltration* case. That is, the MP idea is to simultaneously assess shape characteristics of $\mathcal{G}_t$ based on a multivariate filtering function $F : \mathcal{V}_t \mapsto \mathbb{R}^d$. As a result, e.g., for $d = 2$ and a set of nondecreasing thresholds $\{\alpha_i\}_1^m$ and $\{\beta_j\}_1^n$, instead of a single filtration of complexes, we get a bifiltration of complexes $\{\mathbf{C}_t^{\alpha_i,\beta_j} \mid 1 \leq i \leq m, 1 \leq j \leq n\}$ such that if $\beta_k < \beta_l$, then $\mathbf{C}_t^{\alpha_i,\beta_k} \hookrightarrow \mathbf{C}_t^{\alpha_i,\beta_l}$ and if $\alpha_i < \alpha_j$, then $\mathbf{C}_t^{\alpha_i,\beta_k} \hookrightarrow \mathbf{C}_t^{\alpha_j,\beta_k}$. Finally, this bifiltration induces a bigraded MP module $\{H_k(\mathbf{C}_t^{\alpha_i,\beta_k})\}$, where $H_k$ is the $k^{th}$ homology group. Inspired by Beltramo et al. (2021); Coskunuzer et al. (2021), we propose a new time-aware MP summary, namely, a *Dynamic Euler-Poincaré Surface*.

**Definition 3.1** (Dynamic Euler-Poincaré Surface). Let $\{\mathcal{G}_t\}_{t=1}^T$ be a series of time-varying graphs. Let $F = (f, g)$ be a multivariate filtering function $F : \mathcal{V}_t \mapsto \mathbb{R}^2$ with thresholds $\mathcal{I} = \{(\alpha_i, \beta_j) \mid 1 \leq i \leq m, 1 \leq j \leq n\}$. Let $\mathbf{C}_t^{\alpha_i,\beta_j}$ be the clique complex of the induced subgraph $\mathcal{G}_t^{\alpha_i,\beta_j} = F^{-1}((-\infty, \alpha_i] \times (-\infty, \beta_j])$, $t = 1, 2, \ldots, T$ and $\chi$ be the Euler–Poincaré characteristic. Then, a

sequence of time-evolving $m \times n$-matrices $\{\mathbb{E}^t\}_{t=1}^T$ such that $\mathbb{E}_{ij}^t = \chi(\mathbf{C}_t^{\alpha_i, \beta_j})$ for $1 \leq i \leq m$, $1 \leq j \leq n$ is called *Dynamic Euler-Poincaré Surface* (DEPS). (Figure 2 in Appendix C shows a toy example how DEPS is computed.)

**Theoretical Guarantees of DEPS** Consider two graphs $\mathcal{G}^+$ and $\mathcal{G}^-$, where time index $t$ is suppressed for brevity. Let $F : \mathcal{V}^{\pm} \mapsto \mathbb{R}^2$ be a multivariate filtering function with thresholds $\mathcal{I} = \{(\alpha_i, \beta_j) \mid 1 \leq i \leq m, 1 \leq j \leq n\}$. Let $\mathbf{C}^{\pm}$ be the clique complexes of $\mathcal{G}^{\pm}$, and let $\widehat{\mathbf{C}}^{\pm} = \{\mathbf{C}_{ij}^+\}$ be the bifiltration induced by $(\mathbf{C}^{\pm}, F, \mathcal{I})$ as before. Let $\mathbb{E}^{\pm}$ be the corresponding Euler-Poincaré Surfaces (i.e., $m \times n$ matrices). Then, set $\|\mathbb{E}^+ - \mathbb{E}^-\|_{1,1} = \sum_{i=1}^m \sum_{j=1}^n |\mathbb{E}_{ij}^+ - \mathbb{E}_{ij}^-|$ as the distance between $\mathbb{E}^+$ and $\mathbb{E}^-$, where $\|\cdot\|_{1,1}$ is the vectorized $L_1$ matrix norm.

We now introduce an $L_1$-based MP metric instead of using $L_\infty$-based metrics like, e.g., matching or interleaving, due to the nature of our summaries $\mathbb{E}^{\pm}$ (see Remark C.1 in Appendix C). Let $\mathcal{D}_k^f(\mathbf{C}^{\pm})$ and $\mathcal{D}_k^g(\mathbf{C}^{\pm})$ be the $k^{th}$ single parameter persistence diagrams (PDs) of $\mathbf{C}^{\pm}$ for filtrations induced by functions $f, g : \mathcal{V}^{\pm} \mapsto \mathbb{R}$, respectively (see Appendix B). Let $\mathbf{C}_{i*}^{\pm}$ and $\mathbf{C}_{*j}^{\pm}$ be clique complexes corresponding to $\mathcal{G}_{i*}^{\pm} = f^{-1}((-\infty, \alpha_i])$ and $\mathcal{G}_{*j}^{\pm} = g^{-1}((-\infty, \beta_j])$. Define the $i^{th}$ column distance for the $k^{th}$ PDs as $\mathfrak{D}_{i*}^k(\widehat{\mathbf{C}}^+, \widehat{\mathbf{C}}^-) = \mathcal{W}_1(\mathcal{D}_k^g(\mathbf{C}_{i*}^+), \mathcal{D}_k^g(\mathbf{C}_{i*}^-))$, where $\mathcal{W}_1$ is the Wasserstein-1 distance. Similarly, the $j^{th}$ row distance for $k^{th}$ PDs is $\mathfrak{D}_{*j}^k(\widehat{\mathbf{C}}^+, \widehat{\mathbf{C}}^-) = \mathcal{W}_1(\mathcal{D}_k^f(\mathbf{C}_{*j}^+), \mathcal{D}_k^f(\mathbf{C}_{*j}^-))$.

**Definition 3.2** (Weak $L_1$-metric for Multipersistence). The weak $L_1$-metric between $\widehat{\mathbf{C}}^{\pm}$ is

$$\mathfrak{D}(\widehat{\mathbf{C}}^+, \widehat{\mathbf{C}}^-) = \max\left\{\mathfrak{D}_c(\widehat{\mathbf{C}}^+, \widehat{\mathbf{C}}^-), \mathfrak{D}_r(\widehat{\mathbf{C}}^+, \widehat{\mathbf{C}}^-)\right\},$$

s.t. $\mathfrak{D}_c(\widehat{\mathbf{C}}^+, \widehat{\mathbf{C}}^-) = \sum_{k=0}^M \sum_{i=1}^m \mathfrak{D}_{i*}^k(\widehat{\mathbf{C}}^+, \widehat{\mathbf{C}}^-)$ and $\mathfrak{D}_r(\widehat{\mathbf{C}}^+, \widehat{\mathbf{C}}^-) = \sum_{k=0}^M \sum_{j=1}^n \mathfrak{D}_{*j}^k(\widehat{\mathbf{C}}^+, \widehat{\mathbf{C}}^-)$.

Now, we can state our stability result for Euler-Poincaré Surfaces.

**Theorem 3.1.** *Let $\mathcal{G}^{\pm}, F, \widehat{\mathbf{C}}^{\pm}, \mathbb{E}^{\pm}$ be as defined above. Then, the Euler-Poincaré Surfaces are stable with respect to the weak $L_1$-metric, i.e., $\|\mathbb{E}^+ - \mathbb{E}^-\|_{1,1} \leq C \cdot \mathfrak{D}(\widehat{\mathbf{C}}^+, \widehat{\mathbf{C}}^-)$ for some $C > 0$.*

The proof of the theorem is given in Appendix C. This stability result implies that the distances between multiparameter PDs control the distance between the resulting Euler-Poincaré Surfaces. By combining with the stability result for PDs (Cohen-Steiner et al., 2007), one can conclude that the small changes in the MP filtering function $F : \mathcal{V}_t \mapsto \mathbb{R}^d$ or the small changes in the input data can result only in a small change in DEPS surfaces. For further discussion on implications of the stability result, please refer to Remark C.2.

## 4 TIME-AWARE MULTIPERSISTENCE SPATIO-SUPRA GRAPH CONVOLUTIONAL NETWORKS

Given $\mathcal{T}$ historical observations $\boldsymbol{\mathcal{X}}^{\mathcal{T}} = \{\boldsymbol{X}_{t-\mathcal{T}}, \boldsymbol{X}_{t-\mathcal{T}+1}, \ldots, \boldsymbol{X}_{t-1}\}$, the multivariate forecasting model $\mathfrak{F}(\cdot)$ is learned to predict future observations in the next $\mathcal{H} + 1$ time steps, i.e., $\{\boldsymbol{X}_t, \boldsymbol{X}_{t+1}, \ldots, \boldsymbol{X}_{t+\mathcal{H}}\} = \mathfrak{F}(\boldsymbol{X}_{t-\mathcal{T}}, \boldsymbol{X}_{t-\mathcal{T}+1}, \ldots, \boldsymbol{X}_{t-1})$.

**Graph Learning Architecture** The graph representation learning of our MPS2GCNets is build upon GCN. To learn node representation from graph topology and node features, the input of GCN-based approaches contains the adjacency matrix $A_t$ of the original input graph $\mathcal{G}_t$ and the node feature matrix $\boldsymbol{X}_t$. However, the prior knowledge of graph structure (1) might restrict the modeling capacity (i.e., graph edges cannot encode the complex relationships between nodes) and (2) leads to the neglect of neighboring information with high diversity. To avoid these limitations, inspired by the adaptive dependency matrix (Wu et al., 2019; Chen et al., 2021), we learn the normalized self-adaptive adjacency matrix $\boldsymbol{S} = \{s_{uv}\}_{N \times N}$ with the pre-defined "cost" staying in the same node based on the learnable node embedding $\boldsymbol{E}_\phi = (e_{1,\phi}, e_{2,\phi}, \ldots, e_{N,\phi})^\top \in \mathbb{R}^{N \times d_c}$ as

$$s_{uv} = \begin{cases} \frac{\exp(\text{ReLU}(e_{u,\phi} e_{v,\phi}^\top))}{\sum_{v \in \mathcal{V} \setminus \{u\}} \exp(\text{ReLU}(e_{u,\phi} e_{v,\phi}^\top)) + \exp(d_{uu})}, & u \neq v, \\ \frac{d_{uu}}{\sum_{v \in \mathcal{V} \setminus \{u\}} \exp(\text{ReLU}(e_{u,\phi} e_{v,\phi}^\top)) + \exp(d_{uu})}, & u = v. \end{cases}$$

Here hyperparameter $d_{uu}$ is the "cost" of staying in the same node $u$ and $\text{ReLU}(\cdot) = \max(0, \cdot)$ is an activation function, which guarantees $s_{uv} \geq 0$. Since $\sum_v s_{uv} = 1$ and $s_{uv} \geq 0$, we can use $\boldsymbol{S}$ as the normalized Laplacian. Towards more effective and robust learning of both spatial and spectral characteristics, we represent the graph diffusion as a matrix power series. That is, let $\tilde{\boldsymbol{S}}$ be an $N \times N \times K$ Laplacian tensor containing the power series $\{\boldsymbol{I}, \boldsymbol{S}, \ldots, \boldsymbol{S}^{K-1}\}$ of $\boldsymbol{S}$, where $\boldsymbol{I} \in \mathbb{R}^{N \times N}$ is the identity matrix and $K \geq 2$.

**Spatio-Temporal Feature Transformation** Note that spatial and temporal domains contain interdependent but highly heterogeneous types of information. Inspired by statistical factor analysis, we facilitate signal extraction from these disparate informational sources, by mapping the original input feature space into the high-level latent feature space, which can be written as follows

$$\boldsymbol{H}_{i,FT}^{(\ell+1)} = (\boldsymbol{X}^\mathcal{T} \boldsymbol{E}_\phi \boldsymbol{\Theta}_{FT}^{(\ell)})^\top \boldsymbol{U}, \tag{1}$$

where $\boldsymbol{\Theta}_{FT}^{(\ell)} \in \mathbb{R}^{d_c \times P \times Q_{\mathsf{FT}}}$ and $\boldsymbol{U} \in \mathbb{R}^\mathcal{T}$ represent the learnable parameters.

## 4.1 SUPRAGRAPH CONVOLUTIONAL MODULE IN MULTILAYER SUPRA GRAPH

We propose a novel supragraph convolutional module to simultaneously capture spatio-temporal dependencies in dynamic networks. Our key approach is (i) to represent a sequence of time-varying graphs recorded over a sliding time window, as a *multilayer supra graph*, instead of treating each graph snapshot individually, and (ii) to learn the resulting multilayer supra graph with the random walk exploration which allows us to encode the key details of the time-conditioned relationships among nodes and to boost graph convolutions over multiple edge sets.

**Sliding Window Historical Data as Multilayer Supra Graph Network** To increase expressive capability of spatio-temporal representation learning, we propose a novel directed multilayer supra graph. Particularly, we treat each graph within a sliding window $\mathcal{G}^\mathcal{T} = \{\mathcal{G}_{t-\mathcal{T}}, \mathcal{G}_{t-\mathcal{T}+1}, \ldots, \mathcal{G}_{t-1}\} \in \mathbb{R}^{N \times N \times \mathcal{T}}$ as a layer within a directed $\mathcal{T}$-layered network. (Here $N$ is the number of nodes in each graph.) Since information in real world spatio-temporal processes can propagate only from past to present to future, we consider *directed* multilayer supra graph as abstraction for dynamic knowledge representation. That is, we assume that the information (e.g., spatial features) is shared between layers $t_a$ and $t_b$, whenever $t_b > t_a$, and time stamps $t_a, t_b \in \{t - \mathcal{T}, \ldots, t - 1\}$. More specifically, we propose a strategy to create such kind of information propagation channels by adding directed virtual edges (i.e., interlayer edges) between every node in the layer $t_a$ and its counterparts in other "future" layers $t_b$ $(t_b > t_a)$.

**Definition 4.1.** A directed multilayer supra graph (DMSG) is a tuple, defined as: $\text{DMSG} = (\mathcal{G}_{t-\mathcal{T}}, \ldots, \mathcal{G}_{t-1}, \text{IM}^{t_a t_b})$, where $\mathcal{G}_{t_a} = (\mathcal{V}_{t_a}, \mathcal{E}_{t_a})$, $t_a \in \{t - \mathcal{T}, \ldots, t - 1\}$ are network layers and $\text{IM}^{t_a t_b}$ (Identity Mapping) is an $N \times N$ matrix of node mappings, with $\text{IM}_{ij}^{t_a t_b} : v_i^{t_a} \times v_j^{t_b} \mapsto [0, 1]$. That is, $\text{IM}_{ii}^{t_a t_b}$ is the identity mapping between node $v_i$ in the "past" layer $t_a$ and the "future" layer $t_b$ $(t_b > t_a)$: $\text{IM}_{ii}^{t_a t_b} = 1$.

The corresponding ($r$-th power) supra-adjacency matrix $\boldsymbol{W} = \{w_{uv}\}_{N\mathcal{T} \times N\mathcal{T}}$ for DMSG is then

$$w_{uv}^{ab} = \begin{cases} (s_{uv}^{t_a t_a})^r, & t_a = t_b \text{ and } |u - v| \bmod N \neq 0, \\ (d_{uu}^{t_a t_a})^r, & t_a = t_b \text{ and } |u - v| \bmod N = 0, \\ d_{uu}^{t_a t_b}, & t_a \neq t_b \text{ and } |u - v| \bmod N = 0, \end{cases} \tag{2}$$

where $(s_{uv}^{t_a t_a})^r$ is the $r$-th power of $\boldsymbol{S}^{t_a t_a}$ which encodes the $r$-th step of a random walk among nodes $u$ and $v$ in layer $t_a$, $(d_{uu}^{t_a t_a})^r$ is the "cost" of staying in the same node $u$ and in the same layer $t_a$ after $r$ random walk steps, and $d_{uu}^{t_a t_b}$ is the "cost" of jumping from the current node $u$ in layer $t_a$ to node $u$ in layer $t_b$. Finally, the generalized ($r$-th power) supra-Laplacian $\boldsymbol{\mathcal{L}}_{Sup} \in \mathbb{R}^{N\mathcal{T} \times N\mathcal{T}}$ for DMSG is then

$$\boldsymbol{\mathcal{L}}_{Sup} = \begin{pmatrix} \boldsymbol{D}^{11}\boldsymbol{I}+(\boldsymbol{S}^{11})^r & \boldsymbol{D}^{12}\boldsymbol{I} & \cdots & \boldsymbol{D}^{1\mathcal{T}}\boldsymbol{I} \\ \boldsymbol{0} & \boldsymbol{D}^{22}\boldsymbol{I}+(\boldsymbol{S}^{22})^r & \cdots & \boldsymbol{D}^{2\mathcal{T}}\boldsymbol{I} \\ \vdots & \vdots & \ddots & \vdots \\ \boldsymbol{0} & \boldsymbol{0} & \cdots & \boldsymbol{D}^{\mathcal{T}\mathcal{T}}\boldsymbol{I}+(\boldsymbol{S}^{\mathcal{T}\mathcal{T}})^r \end{pmatrix}. \tag{3}$$

**Supragraph Diffusion Convolutional Layer** Armed with the ($r$-th power) supra-Laplacian in (3), we now build a supragraph diffusion convolutional layer to encode both the graph structure and node

features from the temporal domain. That is, embedded node features are updated by message passing and aggregation via intralayer and interlayer diffusion. The supragraph diffusion convolutional layer is formulated as

$$\boldsymbol{H}_{i,Sup}^{(\ell+1)} = (\boldsymbol{\mathcal{L}}_{Sup}\boldsymbol{H}_{i,Sup}^{(\ell)})^{\top}\boldsymbol{E}_{\phi}\boldsymbol{\Theta}_{Sup}^{(\ell)}, \tag{4}$$

where $\boldsymbol{H}_{i,Sup}^{(\ell)} \in \mathbb{R}^{N\mathcal{T}\times P}$ and $\boldsymbol{H}_{i,Sup}^{(\ell+1)} \in \mathbb{R}^{N\mathcal{T}\times Q_{\mathsf{Sup}}}$ are the input and output activations for layer $\ell$, $\boldsymbol{H}_{i,Sup}^{(0)} = \boldsymbol{X}^{\mathcal{T}}$, and $\boldsymbol{\Theta}_{Sup}^{(\ell)} \in \mathbb{R}^{d_c\times P\times Q_{\mathsf{Sup}}}$ are the learnable parameters. In this case, the diffusion on layer $t_a$ can extend to a fraction of nodes and propagate information of layer $t_b$ by interaction.

## 4.2 SPATIAL GRAPH CONVOLUTIONAL LAYER

To aggregate the features of each node with its multi-hops neighbourhoods to generate node embeddings, we now turn to constructing the spatial graph convolution via

$$\boldsymbol{H}_{i,Spa}^{(\ell+1)} = (\tilde{\boldsymbol{S}}\boldsymbol{H}_{i,Spa}^{(\ell)})^{\top}\boldsymbol{E}_{\phi}\boldsymbol{\Theta}_{Spa}^{(\ell)}, \tag{5}$$

where $\boldsymbol{\Theta}_{Spa}^{(\ell)} \in \mathbb{R}^{d_c\times K\times P\times Q_{\mathsf{Spa}}}$ is the trainable weight, $\tilde{\boldsymbol{S}}$ is the Laplacian tensor, $\boldsymbol{H}_{i,Spa}^{(\ell)} \in \mathbb{R}^{N\times P}$ and $\boldsymbol{H}_{i,Spa}^{(\ell+1)} \in \mathbb{R}^{N\times Q_{\mathsf{Spa}}}$ are the node representations at the $\ell$-th layer and $(\ell+1)$-th layer, respectively, and $\boldsymbol{H}_{i,Spa}^{(0)} = \boldsymbol{X}_i$, i.e., the node features at time step $i$. To reduce computational costs, we use weight sharing matrix factorization instead of matrix entry assignments. As such, we timely update the latest state of variables with back propagation algorithm and reduce the risk of over-fitting.

## 4.3 LEARNING TIME-AWARE MULTIPERSISTENCE REPRESENTATION OF TOPOLOGICAL FEATURES

To fully utilize information delivered by the MP representation mechanism, we propose a CNN-inspired Time-Aware Multipersistence Euler-Poincaré Surface Representation Learning (DEPSRL) module which employs the CNN base model $f_{\theta}$ (including convolutional and pooling layers) and readout layer. In summary, the proposed DEPSRL module offers the following unique innovations: (i) extracting features from the global information, (ii) learning the relationship from DEPS sequences pre and post (i.e., treating the fixed-size sliding window as multi-channels), and (iii) aggregating topological features to make a fixed size representation. The summarized output feature of DEPSRL module is given by

$$\boldsymbol{\Phi}_{i,TAMP}^{(\ell+1)} = \oplus(f_{\mathsf{GAP}}(f_{\theta_1}(\{\mathbb{E}^i\}_{i=1}^{\mathcal{T}})), f_{\mathsf{GMP}}(f_{\theta_2}(\{\mathbb{E}^i\}_{i=1}^{\mathcal{T}}))), \tag{6}$$

where $f_{\mathsf{GAP}}(\cdot)$, $f_{\mathsf{GMP}}(\cdot)$, and $f_{\theta_j}$, $j = \{1, 2\}$, are global average pooling, global max pooling, and $j$-th CNN based model, respectively, and $\oplus$ denotes concatenation, where the $f_{\mathsf{GAP}}(\cdot)$ generates summarized feature for each channel. We find that $f_{\mathsf{GMP}}(\cdot)$ strengthens the representation learning of the time-conditioned multipersistence features. The matrix $\boldsymbol{\Phi}_{i,TAMP}^{(\ell+1)} \in \mathbb{R}^{Q_{\mathsf{TAMP}}}$ is the output of the DEPSRL module. To further learn and fuse multiple latent representations form different views (i.e., spatial information, spatio-temporal correlations, and persistent topological features), we combine these embeddings to obtain the final embedding

$$\boldsymbol{Z}_{i,out}^{(\ell+1)} = \mathscr{F}(\boldsymbol{H}_{i,Spa}^{(\ell+1)}, \boldsymbol{\Upsilon}_i^{(\ell+1)}, \boldsymbol{H}_{i,FT}^{(\ell+1)}). \tag{7}$$

Here $\boldsymbol{\Upsilon}_i^{(\ell+1)} = (1/\mathcal{T}\sum\tilde{\boldsymbol{H}}_{i,Sup}^{(\ell+1)})\boldsymbol{\Phi}_{i,TAMP}^{(\ell+1)}$ integrates global topological and global spatio-temporal information (i.e., we reshape $\boldsymbol{H}_{i,Sup}^{(\ell+1)} \in \mathbb{R}^{N\mathcal{T}\times Q_{\mathsf{Sup}}}$ to $\tilde{\boldsymbol{H}}_{i,Sup}^{(\ell+1)} \in \mathbb{R}^{N\times\mathcal{T}\times Q_{\mathsf{Sup}}}$ and then average $\tilde{\boldsymbol{H}}_{i,Sup}^{(\ell+1)}$ over temporal dimension), $\mathscr{F}(\cdot,\cdot,\cdot)$ is a dimension-wise concatenation function on embeddings along the output dimension, and $\sum_{i\in\{\mathsf{Spa,Sup,FT}\}} Q_i = Q_{\mathsf{out}}$. DEPSRL also allows for input of $\mathbb{E}$ from different bifiltrations. For more details, please refer to Appendix D.2.

## 4.4 MODELING SPATIO-TEMPORAL DYNAMICS

To encode spatio-temporal correlations among time series and get hidden state of nodes at a future timestamp, we put the final embedding $\boldsymbol{Z}_{i,out}^{(\ell+1)}$ into Gated Recurrent Units (GRU). The forward

propagation equations of GRU are as follows

$$\Re_i = \varsigma\left(\boldsymbol{W}_\Re\left[\boldsymbol{\Xi}_{i-1}, \boldsymbol{Z}_{i,out}\right] + \boldsymbol{b}_\Re\right), \qquad \Im_i = \varsigma\left(\boldsymbol{W}_\Im\left[\boldsymbol{\Xi}_{i-1}, \boldsymbol{Z}_{i,out}\right] + \boldsymbol{b}_\Im\right),$$
$$\tilde{\boldsymbol{\Xi}}_i = \tanh\left(\boldsymbol{W}_{\tilde{\Xi}}\left[\Im_i \odot \boldsymbol{\Xi}_{i-1}, \boldsymbol{Z}_{i,out}\right] + \boldsymbol{b}_{\tilde{\Xi}}\right), \qquad \boldsymbol{\Xi}_i = \Re_i \odot \boldsymbol{\Xi}_{i-1} + (1 - \Re_i) \odot \tilde{\boldsymbol{\Xi}}_i, \tag{8}$$

where $\varsigma(\cdot)$ is an activation function which is ReLU in our case, $\odot$ is the elementwise product, $\Re_i$ is the update gate and $\Im_i$ is the reset gate. $\boldsymbol{b}_\Re$, $\boldsymbol{b}_\Im$, $\boldsymbol{b}_{\tilde{\Xi}}$, $\boldsymbol{W}_\Re$, $\boldsymbol{W}_\Im$, and $\boldsymbol{W}_{\tilde{\Xi}}$ are learnable parameters. $\left[\boldsymbol{\Xi}_{i-1}, \boldsymbol{Z}_{i,out}\right]$ and $\boldsymbol{\Xi}_i$ are the input and output of GRU model, respectively. Now we get $\tilde{\boldsymbol{\Xi}}_i$ containing the structure, spatio-temporal, and topological information.

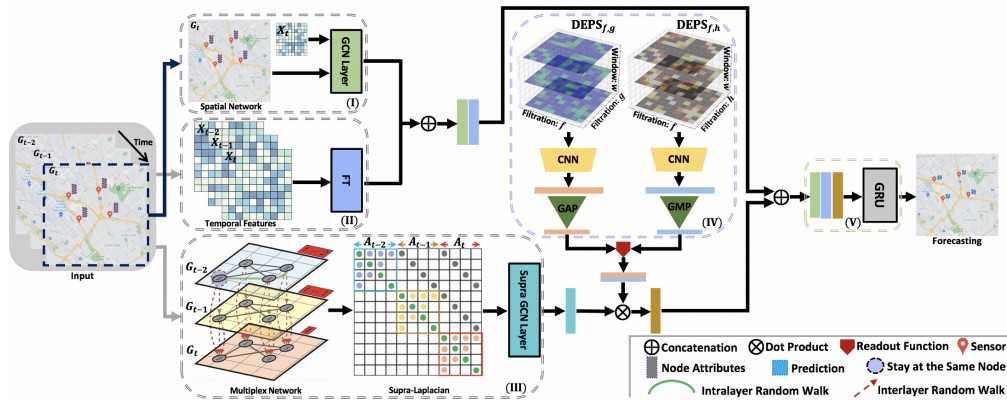

Figure 1: TAMP-S2GCNets consists of 5 components: (I) Spatial graph convolutional layer on $\mathcal{G}_t$ extracts spatial information at time $t$ (see Eq. 5). (II) Feature transformation (FT) learns representation of the spatio-temporal data $\boldsymbol{\mathcal{X}}^{\mathcal{T}}$ over a sliding window of size $\mathcal{T}$ (see Eq. 1). (III) Supragraph convolutional module captures joint spatio-temporal dependencies in $\boldsymbol{\mathcal{X}}^{\mathcal{T}}$ (see Eq. 4). (IV) Detailed architecture of the DEPSRL module (see Eqs. 6 and 7), where the DEPS with different types of multifiltrations can be learned by a CNN base model and global pooling mechanism. Here DEPS allows us to learn the intrinsic graph structure both across multiple geometric dimensions and across time. (V) The learned spatio-temporal dependencies are passed into GRU layer (see Eq. 8) for multi-step forecasting. Note that the whole model can be trained in an end-to-end fashion.

## 5 EXPERIMENTS

**Data Description** We use three different types of datasets to examine the performance of the proposed TAMP-S2GCNets on dynamic networks: (i) The transportation and traffic data of California state from the freeway Performance Measurement System (PeMS). We use three well-studied traffic networks from literature: PeMSD3, PeMSD4 and PeMSD8 (Chen et al., 2001), (ii) Digital transactions between users in the Ethereum blockchain network. We extract dynamic networks from three token assets: Bytom, Decentraland and Golem (Li et al., 2020), and (iii) The spread of coronavirus disease COVID-19 at county-level in states of California (CA) and Texas (TX). More details on each dataset and parameter settings are provided in Appendix D.1.

**Experimental Settings** We compare the presented TAMP-S2GCNets with 13 state-of-the-art baselines (see more details of baselines in Appendix D.2) on three evaluation metrics, i.e, Mean Absolute Error (MAE), Root Mean Squared Error (RMSE), and Mean Absolute Percentage Error (MAPE). For transportation networks, following Cao et al. (2020), we (i) split PeMSD3, PeMSD4, and PeMSD8 datasets into a training set (60%), validation set (20%), and test set (20%) in chronological order and (ii) use the one hour historical data to the next 15 minutes data. Following Chen et al. (2021), we (i) split Bytom, Decentraland, and Golem token networks into training set (80%) and test set (20%) in chronological order and (ii) use 7 days historical data to predict future 7 days data. For COVID-19 datasets, in our experiments, we use daily data of 11 months of 2020, from February 1 to December 31, and split the graph signals into training set, first 80% of days (268 days), and test set, last 20% of days (67 days). Further specifics on experimental setup are provided in Appendix D.2. The best results are in **bold** font and the results with *dotted underline* are the best performance achieved by the runner-ups. We also perform a one-sided two-sample $t$-test between the best result

and the best performance achieved by the runner-up, where *, **, *** denote significant, statistically significant, highly statistically significant results, respectively. More detailed experiments on all datasets are in Appendix E. The data and code implementation are available at https://www.dropbox.com/sh/n0ajd5l0tdeyb80/AABGn-ejfV1YtRwjf_L0AOsNa?dl=0.

**Computational Complexity** Although, in general, MP is computationally costly, in our case, to get the DEPS summary, we *do not need* to compute PDs as in other MP approaches, but only Betti numbers of each filtration cell. In particular, while a computational cost for obtaining $k^{th}$-PD for a graph is $\mathcal{O}(\mathcal{M}_k^3)$, where $\mathcal{M}_k$ is the number $k$-simplices (Otter et al., 2017), obtaining Euler Characteristics by sparse matrix methods has computational complexity of $\mathcal{O}(\mathcal{M}_0 + \mathcal{M}_1 + \mathcal{M}_2)$ (Edelsbrunner & Parsa, 2014).

## 5.1 EXPERIMENTAL RESULTS

**Transportation Traffic Flow** Table 1 shows the performance comparison among 13 state-of-the-art baselines on PeMSD3, PeMSD4, and PeMSD8 for multi-step traffic flow forecasting. Our TAMP-S2GCNets consistently outperforms baseline models on all 3 datasets except for PeMSD3, which underscore effectiveness of TAMP-S2GCNets in time series forecasting tasks. On PeMSD3, our TAMP-S2GCNets achieves the best MAE and MAPE, and has an average 2.75% relative gain, compared to the runner-up. The average relative gains of TAMP-S2GCNets over the runner-ups in MAE and MAPE are 2.78% and 2.19% on the 3 datasets, respectively.

**Ethereum Blockchain Prices** Table 3 (**left**) summarizes the comparison results in MAPE on 3 Ethereum token networks (i.e., Bytom, Decentrland, and Golen). Table 3 indicates that TAMP-S2GCNets is always better than baselines for all dynamic Ethereum networks. We find that, even compared to the baseline which also integrates topological information (i.e., Z-GCNETs), TAMP-S2GCNets is highly competitive, implying that the time-aware MP representation with DEPS is capable of better encoding time-conditioned information than the zigzag idea based on one filtration.

**COVID-19 Hospitalizations** Table 2 shows results on 15-day ahead forecasting of COVID-19 hospitalizations in the U.S. states of California (CA) and Texas (TX) at a county level basis (RMSE is aggregated over each state). For the sake of room, here we only display the top runner models. TAMP-S2GCNets yields significantly better forecasting performance, with relative gains of 1.5%-24.5% (in CA) and 5.5%-46.7% (in TX).

Table 1: Forecasting performance on PeMSD3, PeMSD4 and PeMSD8, along with results on statistical significance. Standard deviations are suppressed for the sake of room and are reported in Table 8 in Appendix E.

| Model | PeMSD3 | | | PeMSD4 | | | PeMSD8 | | |
|---|---|---|---|---|---|---|---|---|---|
| | MAE | RMSE | MAPE (%) | MAE | RMSE | MAPE (%) | MAE | RMSE | MAPE (%) |
| FC-LSTM (Sutskever et al., 2014) | 21.33 | 35.11 | 23.33 | 27.14 | 41.59 | 18.20 | 22.20 | 34.06 | 14.20 |
| SFM (Zhang et al., 2017) | 17.67 | 30.01 | 18.33 | 24.36 | 37.10 | 17.20 | 16.01 | 27.41 | 10.40 |
| N-BEATS (Oreshkin et al., 2019) | 18.45 | 31.23 | 18.35 | 25.56 | 39.90 | 17.18 | 19.48 | 28.32 | 13.50 |
| DCRNN (Li et al., 2018) | 18.18 | 30.31 | 18.91 | 24.70 | 38.12 | 17.12 | 17.86 | 27.83 | 11.45 |
| LSTNet (Lai et al., 2018) | 19.07 | 29.67 | 17.73 | 24.04 | 37.38 | 17.01 | 20.26 | 31.96 | 11.30 |
| STGCN (Yu et al., 2018) | 17.49 | 30.12 | 17.15 | 22.70 | 35.50 | 14.59 | 18.02 | 27.83 | 11.40 |
| TCN (Bai et al., 2018) | 18.23 | 25.04 | 19.44 | 26.31 | 36.11 | 15.62 | 15.93 | 25.69 | 16.50 |
| DeepState (Rangapuram et al., 2018) | 15.59 | *20.21 | 18.69 | 26.50 | 33.00 | 15.40 | 19.34 | 27.18 | 16.00 |
| GraphWaveNet (Wu et al., 2019) | 19.85 | 32.94 | 19.31 | 26.85 | 39.70 | 17.29 | 19.13 | 28.16 | 12.68 |
| DeepGLO (Sen et al., 2019) | 17.25 | 23.25 | 19.27 | 25.45 | 35.90 | 12.20 | 15.12 | 25.22 | 13.20 |
| AGCRN (Bai et al., 2020) | 14.22 | 24.03 | 13.89 | 17.78 | 29.17 | 11.79 | 14.59 | 23.06 | 9.29 |
| Z-GCNETs (Chen et al., 2021) | 14.20 | 25.29 | 13.88 | 18.05 | 29.08 | 11.79 | 14.52 | 23.00 | 9.28 |
| StemGNN (Cao et al., 2020) | 14.32 | 21.64 | 16.24 | 20.20 | 31.83 | 12.00 | 15.83 | 24.93 | 9.26 |
| **TAMP-S2GCNets (ours)** | **13.91** | 23.77 | **13.40** | 17.58 | **28.56 | *11.01 | 13.77 | **21.70 | 8.99 |

## 5.2 ABLATION STUDIES

**Contribution of TAMP-S2GCNets components** We now conduct the ablation studies on PeMSD4 and PeMSD8 to evaluate contribution of different components of our framework. (Ablation results on Golem and CA are in Table 9 in Appendix E.)

We compare our TAMP-S2GCNets with 4 ablated variants, i.e., (i) TAMP-S2GCNets without DEPSRL module (w/o DEPSRL module), (ii) TAMP-S2GCNets without spatial graph convolutional layer (w/o $GCN_{Spa}$), (iii) TAMP-S2GCNets without supragraph convolutional module (w/o $GCN_{Sup}$), and (iv) TAMP-S2GCNets without FT (w/o FT). As Table 3 (**right panel**) shows, ablating each of above causes the

Table 2: 15-day ahead forecasting results (RMSE) on COVID-19 hospitalizations in CA and TX.

| Model | CA | TX |
|---|---|---|
| DCRNN (Li et al., 2018) | 492.10±2.96 | 90.47±2.28 |
| AGCRN (Bai et al., 2020) | 448.27±2.78 | 52.96±3.92 |
| StemGNN (Cao et al., 2020) | 377.25±3.91 | 51.00±2.60 |
| **TAMP-S2GCNets (ours)** | *371.60±2.68 | *48.21±3.17 |

performance drops sharply in comparison with our full TAMP-S2GCNets model. Especially, on PeMSD4, the supragraph convolutional module significantly improves the results as it simultaneously captures the spatial and temporal information. In addition, it is evident that DEPSRL module enhances the topological information learning ability in spatio-temporal domain, i.e., TAMP-S2GCNets outperforms TAMP-S2GCNets w/o DEPSRL module with an average relative gain 3.92% on RMSE over PeMSD4 and PeMSD8. Besides, as expected, taking the advantages of $GCN_{Spa}$ and FT, we improve the performance via capturing the graph structure in the spatial dimension and consolidating the processing of spatio-temporal correlations between node attributes, respectively.

Table 3: Forecasting performance (MAPE in %) on Ethereum networks (**left panel**) and the TAMP-S2GCNets ablation study on PeMSD4 and PeMSD8 (**right panel**).

| Model | Bytom | Decentraland | Golem |
|---|---|---|---|
| DCRNN (Li et al., 2018) | 35.36±1.18 | 27.69±1.77 | 23.15±1.91 |
| STGCN (Yu et al., 2018) | 37.33±1.06 | 28.22±1.69 | 23.68±2.31 |
| GraphWaveNet (Wu et al., 2019) | 39.18±0.96 | 37.67±1.76 | 28.89±2.34 |
| AGCRN (Bai et al., 2020) | 34.46±1.37 | 26.75±1.51 | 22.83±1.91 |
| Z-GCNETs (Chen et al., 2021) | 31.04±0.78 | 23.81±2.43 | 22.32±1.42 |
| StemGNN (Cao et al., 2020) | 34.91±1.04 | 28.37±1.96 | 22.50±2.01 |
| **TAMP-S2GCNets (ours)** | *29.26±1.06 | ***19.89±1.49 | **20.10±2.30 |

| | Architecture | MAE | RMSE | MAPE |
|---|---|---|---|---|
| | **TAMP-S2GCNets** | **17.58** | **28.56** | **11.01** |
| | W/o DEPSRL | 17.89 | ***29.99 | *11.07 |
| PeMSD4 | W/o $GCN_{Spa}$ | **19.41 | ***30.90 | ***11.21 |
| | W/o $GCN_{Sup}$ | ***20.61 | ***31.82 | ***12.64 |
| | W/o FT | *18.30 | ***29.65 | ***12.29 |
| | **TAMP-S2GCNets** | **13.77** | **21.70** | **8.99** |
| | W/o DEPSRL | ***14.28 | **22.39 | 9.32 |
| PeMSD8 | W/o $GCN_{Spa}$ | 14.16 | **22.29 | 9.40 |
| | W/o $GCN_{Sup}$ | 13.99 | *21.91 | 9.07 |
| | W/o FT | ***14.36 | ***22.60 | 9.26 |

Table 4: MAPE (%) (standard deviation) of TAMP-S2GCNets with single- and multifiltrations (**left panel**) and MAPE (%) and computation time for TAMP-S2GCNets with our DEPS, MP-I of Carrière & Blumberg (2020) and MP-L of Vipond (2020) (**right panel**).

| TAMP-S2GCNets on Bytom (Left) | | TAMP-S2GCNets on Bytom (Right) | | |
|---|---|---|---|---|
| **Single filtration** | **Multifiltration** | **MP summary** | **MAPE (%)** | **Running time (s)** |
| - | Deg & Btwns: 30.02±1.05 | MP-I (Carrière & Blumberg, 2020) | 33.13 | 401.80 |
| Deg: 30.56±1.08 | Btwns & $Power_{Trns}$: **29.26±0.96** | MP-L (Vipond, 2020) | 32.19 | 517.11 |
| Btwns: 30.80±1.70 | Btwns & $Power_{Vol}$: 29.27±0.78 | $DEPS_{Deg\ \&\ Btwns}$ | 30.02 | 47.67 |
| $Power_{Trns}$: 31.04±1.90 | Deg & $Power_{Trns}$: 29.86±1.05 | $DEPS_{Deg\ \&\ Power_{Trns}}$ | 29.86 | 39.46 |
| $Power_{Vol}$: 30.79±1.61 | Deg & $Power_{Vol}$: 29.41±1.06 | $DEPS_{Btwns\ \&\ Power_{Trns}}$ | **29.26** | **29.84** |

**Contribution of Single vs. Multifiltration Persistence** We consider sublevel filtrations based on node degree (Deg), betweenness (Btwns), and graph diameter (power filtration) with edge weights induced by number of transactions ($Power_{Trns}$) and volume of transactions ($Power_{Vol}$). As Table 4 (**left**) shows, MP always outperforms single filtrations by a significant margin, both in terms of prediction accuracy and variability. The results confirm our premise that MP is able (i) to capture hidden time dependencies of high dimensional time-varying objects which are inaccessible with one-parameter PH and (ii) to yield substantially more stable feature maps in dynamic scenarios.

**Contribution of DEPS vs. the Existing MP Summaries** Finally, we evaluate performance of the new time-aware MP summary, i.e., DEPS, in comparison to the MP summaries based on the slicing argument, namely, MP-I of Carrière & Blumberg (2020) and MP-L of Vipond (2020) as representation input to TAMP-S2GCNets. As Table 4 (**right panel**) shows, comparing to MP-I and MP-L, DEPS yields 9%-12% improvement in MAPE, and at least a **10 times decrease** in computational time (29.84s for DEPS vs. 401.80s for MP-I and 517.11s for MP-L). Such substantial differences in computational costs are explained by the need of the slicing-based MP summaries to search for the most suitable one-parameter PH representation of MP. In turn, DEPS is based on a

point-wise representation argument in linear algebra, and its high computational efficiency makes DEPS the preferred choice for time-conditioned topological representation learning (see the running time in Table 4). Since computational costs remain the major roadblock for MP applications, our DEPS approach shows that MP tools based on scalable sparse matrix algorithms appear to be the most promising direction for integration of MP into machine learning tasks.

## 6 CONCLUSION

We have explored utility of MP to enhance knowledge representation mechanisms within the time-aware DL paradigm. The developed TAMP-S2GCNets model is shown to yield highly competitive forecasting performance on a wide range of datasets, with much lower computational costs. In the future we plan to explore combination of MP with the zigzag persistence and to investigate asymptotic distributional properties of point-wise MP invariants.

## ETHICS STATEMENT

We do not anticipate any negative ethical implications of the proposed methodology. In turn, we believe that introduction of GCNs tools, coupled with multiparameter topological approaches, into biosurveillance opens a new path for more accurate, timely, and robust tracking of infectious diseases with high virulence such as COVID-19. In particular, multiparameter persistence allows for extracting the most salient topological features along multiple geometric dimensions and, as such, can be especially valuable to address disease dynamics as a function of multiple highly heterogeneous variables, e.g., socio-demographic, socio-environmental, and socio-economic factors. In turn, GCNs and, more generally, geometric deep learning allow for capturing sophisticated nonlinear spatio-temporal dependencies among factors which contribute to the disease spread. Hence, we postulate that in the next few years we can expect to see a new set of spatio-temporal biosurveillance artificial intelligence algorithms, based on a combination of geometric deep learning, multiparameter persistence, and more generally, tools of topological data analysis.

## REPRODUCIBILITY STATEMENT

The source code for the experiments can be accessed under `https://github.com/tamps2gcnets/TAMP_S2GCNets.git`.

## ACKNOWLEDGEMENTS

This work is sponsored by the National Science Foundation under award numbers ECCS 2039701, INTERN supplement for ECCS 1824716, DMS 1925346 and the Department of the Navy, Office of Naval Research under ONR award number N00014-21-1-2530. Part of this material is also based upon work supported by (while serving at) the National Science Foundation. Any opinions, findings, and conclusions or recommendations expressed in this material are those of the author(s) and do not necessarily reflect the views of the National Science Foundation and/or the Office of Naval Research. The authors are also grateful to the ICLR anonymous reviewers for many insightful suggestions and engaging discussion which improved clarity of the manuscript and highlighted new open research directions.

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

## A  NOTATION

Frequently used notation is summarized in Table 5.

Table 5: The main symbols and definitions in this paper.

| Notation | Definition |
|---|---|
| $\mathcal{G}_t$ | the spatial network at timestamp $t$ |
| $N$ | the number of nodes |
| $M_t$ | the number of edges at timestamp $t$ |
| $\boldsymbol{\mathcal{X}}$ | a sequence of observations on a multivariate variable |
| $\boldsymbol{X}_t$ | node feature matrix at timestamp $t$ |
| $\mathbf{C}$ | simplicial complex |
| $f$ | filtering function for sublevel filtration |
| $M_k$ | number of $k$-simplices |
| $\mathcal{D}_k(\mathbf{C})$ | $k$-dimensional persistence diagram |
| $\chi(\mathbf{C})$ | Euler-Poincaré Characteristics |
| $H_k(\mathbf{C})$ | $k^{th}$ homology group |
| $F(\cdot, \cdot)$ | multivariate filtering function |
| $\mathbb{E}^{\pm}$ | Euler-Poincaré Surfaces |
| $\{\mathbb{E}^t\}_{t=1}^T$ | Dynamic Euler-Poincaré Surface |
| $(\Sigma, \mathcal{A}, \mathbb{P})$ | a probability space |
| $\mathbb{F}$ | a filtration of $\sigma$-fields |
| $\mathcal{W}_1$ | Wasserstein-1 distance |
| $\mathfrak{D}(\cdot, \cdot)$ | the distance function for MP |
| $\mathcal{B}(\cdot)$ | the Betti function |
| $\mathcal{M}_k$ | the number $k$-simplices |
| $F_N$ | the feature dimension |
| $\mathcal{T}$ | the sliding window size |
| $\mathfrak{F}(\cdot)$ | the multivariate forecasting model |
| $\boldsymbol{S}$ | the normalized self-adaptive adjacency matrix |
| $\boldsymbol{E}_\phi$ | the learnable node embedding |
| $\tilde{\boldsymbol{S}}$ | Laplacian tensor |
| $K$ | the length of Laplacian tensor $\tilde{\boldsymbol{S}}$ |
| $\mathcal{L}$ | supra-Laplacian |
| $\boldsymbol{\Phi}$ | the output of DEPSRL module |
| $\mathscr{F}(\cdot, \cdot, \cdot)$ | the dimension-wise concatenation function |

## B  SINGLE PARAMETER PERSISTENT HOMOLOGY AND ITS SUMMARIES

All extracted topological features can be then summarized as a multiset in $\mathbb{R}^2$, called *persistence diagram (PD)*: $\mathcal{D}_k(\mathbf{C}) = \{(b_i, d_i) \in \mathbb{R}^2 \mid b_i < d_i\}$, where birth $b_i$ and death $d_i$ mark the filtration indexes at which the $k$-dimensional topological feature $\rho_i$ appears and disappears, respectively. The farther $(b_i, d_i)$ from the diagonal is (that is, the more persistent $\rho_i$ is), the likelier $\rho_i$ is to contain salient information on $\mathcal{G}_t$. Multiplicity of $(x_b, y_d) \in \mathcal{D}$ is the number of $p$-dimensional topological features ($p$-holes) that are born and die at $x_b$ and $y_b$, respectively. Features on the diagonal of $\mathcal{D}$ have infinite multiplicity.

Another summary which may be particularly suitable in our context of modeling noisy time-evolving objects is the Euler-Poincaré Characteristics (see Adler et al. (2010); Baryshnikov & Ghrist (2010) on the Euler calculus and its applications, particular, in conjunction with sensor networks).

**Definition B.1** (Euler-Poincaré Characteristics). For a given simplicial complex $\mathbf{C}$, Euler-Poincaré Characteristics $\chi(\mathbf{C})$ is defined as the alternating sum of the number of $k$-simplices of $\mathbf{C}$. That is, if $n_k$ is the number of $k$-simplices in $\mathbf{C}$, then $\chi(\mathbf{C}) = \sum_{k=0}^{M}(-1)^k n_k$, where $M$ is the dimension of highest dimensional simplex in $\mathbf{C}$. Note that Euler-Poincaré Characteristics is homotopy invariant,

and an alternative definition is $\chi(\mathbf{C}) = \sum_{k=0}^{M}(-1)^k \mathcal{B}_k(\mathbf{C})$ where $\mathcal{B}_k(\mathbf{C})$ is the $k^{th}$ Betti number of $\mathbf{C}$, i.e. rank of the $k^{th}$ homology group $H_k(\mathbf{C})$ (Ghrist, 2018; Hatcher, 2002).

## C   PROOF OF THEOREM 3.1

In this section, we provide the proof of the stability theorem (Theorem 3.1). First, let us recall the key notations we use. Let $\mathcal{G}^+$ and $\mathcal{G}^-$ be two graphs and let $\mathbf{C}^{\pm}$ be the clique complexes of $\mathcal{G}^{\pm}$ (Ghrist, 2018). Let $F = (f, g)$ be a multivariate filter function, i.e. $F : \mathcal{V}^{\pm} \to \mathbb{R}^2$, where $\mathcal{V}^{\pm}$ is the set of nodes of $\mathcal{G}^{\pm}$, and let $\mathcal{I} = \{(\alpha_i, \beta_j) \mid 1 \leq i \leq m, 1 \leq j \leq n\}$ be the corresponding thresholds for $F = (f, g)$.

Next, we define Wasserstein-$p$ distance among PDs, i.e., the important concept we borrow from the theory of single parameter persistence (Edelsbrunner & Harer, 2010).

**Definition C.1** (Wasserstein-$p$ distance). Let $(\mathbf{C}^{\pm}, f^{\pm}, \mathcal{I}^{\pm})$ be two single parameter filtrations, and $\mathcal{D}_k(\mathbf{C}^{\pm})$ be the corresponding PDs for $k$-cycles (i.e, $k$-dimensional topological features). Let $q_j^+ = (b_j^+, d_j^+) \in \mathcal{D}_k(\mathbf{C}^+)$ be the birth and death times of a $k$-dimensional topological feature $\rho_j$. Then, Wasserstein-$p$ distance between $\mathcal{D}_k(\mathbf{C}^+)$ and $\mathcal{D}_k(\mathbf{C}^-)$ is defined as

$$\mathcal{W}_p(\mathcal{D}_k(\mathbf{C}^+), \mathcal{D}_k(\mathbf{C}^-)) = \inf_{\phi} \left( \sum_{j \in \mathcal{D}(\mathbf{C}^+) \cup \Delta} \|q_j^+ - \phi(q_j^+)\|_{\infty}^p \right)^{\frac{1}{p}}, \quad p \in \mathbb{Z}^+, \tag{9}$$

where $\phi : \mathcal{D}_k(\mathbf{C}^+) \cup \Delta \to \mathcal{D}_k(\mathbf{C}^-) \cup \Delta$ is a bijection (matching) and $\Delta$ is the diagonal set, i.e., $\Delta = \{(b, b) \mid b \in \mathbb{R}\}$, which contains $k$-cycles of infinite multiplicity. With $\Delta$ in both sides, we ensure the existence of these bijections even if the cardinalities $|\{q_j^+\}|$ and $|\{q_l^-\}|$ are different. When $p = \infty$, (9) corresponds to the bottleneck distance, i.e. $\mathcal{W}_{\infty}$. In the proof below, we use $p = 1$, i.e. $\mathcal{W}_1$-distance.

Armed the PD and Wasserstein-$p$ distance concepts, we now turn to the main stability result for the new MP summary. Let $\widehat{\mathbf{C}}^{\pm} = \{\mathbf{C}_{ij}^{\pm}\}$ be the bifiltration associated with $(\mathbf{C}^{\pm}, F, \mathcal{I})$, i.e. $\mathbf{C}_{ij}^{\pm}$ is the clique complex of the induced subgraph $\mathcal{G}_{ij}^{\pm} = F^{-1}((-\infty, \alpha_i] \times (-\infty, \beta_j])$. (Figure 2 illustrates such bifiltration on graphs with degree and eccentricity functions.) Let $\mathbb{E}^{\pm}$ be the the corresponding Euler-Poincaré Surface, i.e. $\mathbb{E}^{\pm}$ is an $m \times n$-matrix with entries $\mathbb{E}_{ij}^{\pm} = \chi(\mathbf{C}_{ij}^{\pm},)$, where $\chi(.)$ is the Euler Characteristics (See Figure 2), and $\mathfrak{D}(\mathbf{C}^+, \mathbf{C}^-)$ be the weak $L_1$-distance as defined in Section 3 of the paper (see also Remark C.1 below for the motivation to use this new metric).

**Proof of Theorem 3.1:** We start from providing the outline of the proof. Notice that Euler-Poincaré Surfaces is the alternating sum of Betti numbers $\{\mathcal{B}_k\}$. Hence, the distance $\|\mathbb{E}^+ - \mathbb{E}^-\|_{1,1}$ is bounded by the difference of the corresponding Betti numbers for $\widehat{\mathbf{C}}^{\pm}$, i.e. Eqs. (10) and (11). By fixing $i$ and $k$, and by taking column distance $\mathfrak{D}_c$ into consideration, our main assertion

$$\|\mathbb{E}^+ - \mathbb{E}^-\|_{1,1} \leq C \cdot \mathfrak{D}(\widehat{\mathbf{C}}^+, \widehat{\mathbf{C}}^-), \quad C > 0$$

then reduces to Eqs. (12) and (13). To relate the difference $|\mathcal{B}(\mathbf{C}_j^+) - \mathcal{B}(\mathbf{C}_j^-)|$ to $\mathcal{W}_1(\mathcal{D}(\mathbf{C}^+), \mathcal{D}(\mathbf{C}^-))$, we observe that the Betti number $\mathcal{B}(\mathbf{C}_j^{\pm})$ is the sum of the number of barcodes in $\mathcal{D}(\mathbf{C}^{\pm})$, containing $\beta_j$, i.e., $\#\{r \mid \beta_j \in [b_r^{\pm}, d_r^{\pm}]\}$ (Eqs. (13) and (14)). Yet, the differences of the number of barcodes containing $\beta_j$ for $\mathbf{C}^+$ and $\mathbf{C}^-$ can be bounded by the Wasserstein-1 distances of $\mathcal{D}(\mathbf{C}^+)$ and $\mathcal{D}(\mathbf{C}^-)$, i.e. Eqs. (15) and (16). Aggregating these terms on indices $i$ and $k$ finishes the proof.

Now, we turn to derivations in details. Recall that $\chi(\mathbf{C}) = \sum_{k=0}^{M}(-1)^k \mathcal{B}_k(\mathbf{C})$, where $\mathcal{B}_k(\mathbf{C})$ is the $k^{th}$ Betti number of $\mathbf{C}$ and $M$ is the maximum homological dimension of $\mathbf{C}$. Then, since $\mathbb{E}_{ij}^{\pm} = \chi(\mathbf{C}_{ij}^{\pm})$, we have $\mathbb{E}_{ij}^{\pm} = \sum_{k=0}^{M} \mathcal{B}_k(\mathbf{C}_{ij}^{\pm})$. Hence, for any $i, j$,

$$|\mathbb{E}_{ij}^+ - \mathbb{E}_{ij}^-| = |\sum_{k=0}^{M}(-1)^k [\mathcal{B}_k(\mathbf{C}_{ij}^+) - \mathcal{B}_k(\mathbf{C}_{ij}^-)]| \leq \sum_{k=0}^{M} |\mathcal{B}_k(\mathbf{C}_{ij}^+) - \mathcal{B}_k(\mathbf{C}_{ij}^-)|. \tag{10}$$

Since $\|\mathbb{E}^+ - \mathbb{E}^-\|_{1,1} = \sum_{i=1}^m \sum_{j=1}^n |\mathbb{E}_{ij}^+ - \mathbb{E}_{ij}^-|$, we have

$$\|\mathbb{E}^+ - \mathbb{E}^-\|_{1,1} \leq \sum_{i=1}^m \sum_{j=1}^n \sum_{k=0}^M |\mathcal{B}_k(\mathbf{C}_{ij}^+) - \mathcal{B}_k(\mathbf{C}_{ij}^-)|. \tag{11}$$

Now, to complete the proof, we need to bound the latter sum by the MP distance $\mathfrak{D}(\widehat{\mathbf{C}}^+, \widehat{\mathbf{C}}^-)$. Recall that by Definition 3.3

$$\mathfrak{D}_c(\widehat{\mathbf{C}}^+, \widehat{\mathbf{C}}^-) = \sum_{k=0}^M \sum_{i=1}^m \mathcal{W}_1(\mathcal{D}_k^g(\mathbf{C}_{i*}^+), \mathcal{D}_k^g(\mathbf{C}_{i*}^-)) \leq \mathfrak{D}(\widehat{\mathbf{C}}^+, \widehat{\mathbf{C}}^-).$$

Then, by Eq. 11, to prove the theorem, it is enough to show that for any $0 \leq k_0 \leq M$ and for any $1 \leq i_0 \leq m$,

$$\sum_{j=1}^n \left|\mathcal{B}_{k_0}(\mathbf{C}_{i_0 j}^+) - \mathcal{B}_{k_0}(\mathbf{C}_{i_0 j}^-)\right| \leq C \cdot \mathcal{W}_1(\mathcal{D}_{k_0}^g(\mathbf{C}_{i_0*}^+), \mathcal{D}_{k_0}^g(\mathbf{C}_{i_0*}^-)), \qquad C > 0. \tag{12}$$

For simplicity, without loss of generality, we fix $1 \leq i_0 \leq m$ and $0 \leq k_0 \leq N$. Also, for sake of notations, we drop the subscripts and superscripts, i.e $\mathbf{C}^\pm = \mathbf{C}_{i_0*}^\pm$ and $\mathbf{C}_j^\pm = \mathbf{C}_{i_0 j}^\pm$, $\mathcal{D}(\mathbf{C}^\pm) = \mathcal{D}_{k_0}^g(\mathbf{C}_{i_0*}^\pm)$ and $\mathcal{B}(.) = \mathcal{B}_{k_0}(.)$. With these simplified notations, Eq. 12 is equivalent to

$$\sum_{j=1}^n \left|\mathcal{B}(\mathbf{C}_j^+) - \mathcal{B}(\mathbf{C}_j^-)\right| \leq C \cdot \mathcal{W}_1(\mathcal{D}(\mathbf{C}^+), \mathcal{D}(\mathbf{C}^-)) \qquad C > 0. \tag{13}$$

To show (13), we need to evaluate the relation between the Betti numbers $\mathcal{B}(\mathbf{C}_j^\pm)$ and the persistence diagrams $\mathcal{D}(\mathbf{C}^\pm)$. Consider $\mathcal{D}(\mathbf{C}) = \{(b_r, d_r) \mid 1 \leq r \leq Q\}$, where $b_r$ represents the birth time of a $k_0$-cycle $\rho_r$ in $\mathbf{C}$, while $d_r$ represents the death time of $\rho_r$. In particular, $\rho_r$ induces a barcode $[b_r, d_r)$. For each such $\rho_r$, define indicator function $\psi_r(t) = \mathcal{I}_{[b_r, d_r)}(t)$, i.e. $\psi_r(t) = 1$ for $t \in [b_r, d_r)$ and $\psi_r(t) = 0$ otherwise. Then, the Betti function of $\mathbf{C}$ is defined as $\mathcal{B} : [\beta_1, \infty) \to \mathbb{N}$ such that $\mathcal{B}(t) = \sum_{r=1}^Q \psi_r(t)$ for $t \in [\beta_1, \infty)$. (Here, $\beta_1$ is the lowest threshold for the filtering function $g$ in the set of thresholds $\mathcal{I} = \{(\alpha_i, \beta_j) \mid 1 \leq i \leq m, 1 \leq j \leq n\}$. Similarly, if we consider row sum $\mathfrak{D}_r$ instead of column sum $\mathfrak{D}_c$, then we would use the other filtering function $f$, and the Betti functions would be defined as $\mathcal{B} : [\alpha_1, \infty) \to \mathbb{N}$.) Hence,

$$\mathcal{B}(\mathbf{C}_j^\pm) = \sum_{r=1}^{Q^\pm} \psi_r^\pm(\beta_j), \quad j \in \mathbb{Z}^+.$$

As a result, we get

$$\sum_{j=1}^n \left|\mathcal{B}(\mathbf{C}_j^+) - \mathcal{B}(\mathbf{C}_j^-)\right| = \sum_{j=1}^n \left|\sum_{r=1}^{Q^+} \psi_r^+(\beta_j) - \sum_{r=1}^{Q^-} \psi_r^-(\beta_j)\right| \tag{14}$$

$$\leq \sum_{j=1}^n \sum_{r=1}^{\widehat{Q}} |\psi_r^+(\beta_j) - \psi_{\phi(r)}^-(\beta_j)|,$$

where $\phi : \mathcal{D}(\mathbf{C}^+) \to \mathcal{D}(\mathbf{C}^-)$ is the best matching in the definition of $\mathcal{W}_1(\mathcal{D}(\mathbf{C}^+), \mathcal{D}(\mathbf{C}^-))$ and $\widehat{Q} = \max\{Q^+, Q^-\}$.

Notice that the contribution of each $|\psi_r^+(\beta_j) - \psi_{\phi(r)}^-(\beta_j)|$ to the sum at $\beta_j$ is less than the number $(\#r)$ of intervals $[b_r^+, d_r^+) \triangle [b_{\phi(r)}^+, b_{\phi(r)}^-)$ containing the interval $[\beta_j, \beta_{j+1})$, where $\triangle$ is the symmetric difference of the sets. Let $K = \min_j(\beta_{j+1} - \beta_j)$. By dividing the length $\|[b_r^+, d_r^+) \triangle [b_{\phi(r)}^+, b_{\phi(r)}^-)\|$ by $K$, we ensure that every such interval contributes at least 1 to the Betti number count. Hence, we arrive at the following key inequality:

$$\sum_{j=1}^n \sum_{r=1}^{\widehat{Q}} |\psi_r^+(\beta_j) - \psi_{\phi(r)}^-(\beta_j)| \leq \frac{1}{K} \cdot \sum_r |b_r^+ - b_{\phi(r)}^-| + |d_r^+ - d_{\phi(r)}^-|. \tag{15}$$

Finally, since

$$\sum_{r=1}^{\widehat{Q}} |b_r^+ - b_{\phi(r)}^-| + |d_r^+ - d_{\phi(r)}^-| \leq 2\mathcal{W}_1(\mathcal{D}(\mathbf{C}^+), \mathcal{D}(\mathbf{C}^-)),$$

we obtain

$$\sum_{j=1}^{n} |\mathcal{B}^+(\beta_j) - \mathcal{B}^-(\beta_j)| \leq \frac{2}{K} \cdot \mathcal{W}_1(\mathcal{D}(\mathbf{C}^+), \mathcal{D}(\mathbf{C}^-)). \tag{16}$$

Since (16) holds for any $1 \leq i_0 \leq m$ and $0 \leq k_0 \leq N$, for $C = 2/K$ we get

$$\sum_{i=1}^{m} \sum_{j=1}^{n} \sum_{k=0}^{M} |\mathcal{B}_k(\mathbf{C}_{ij}^+) - \mathcal{B}_k(\mathbf{C}_{ij}^-)| \leq C \cdot \mathfrak{D}_c(\widehat{\mathbf{C}}^+, \widehat{\mathbf{C}}^-).$$

By Eq. (11), this implies

$$\|\mathbb{E}^+ - \mathbb{E}^-\|_{1,1} \leq C \cdot \mathfrak{D}(\widehat{\mathbf{C}}^+, \widehat{\mathbf{C}}^-),$$

which completes the proof. $\qquad\square$

**Remark C.1 (On the Choice of the Metric).** While interleaving and matching distances are very common and useful for multipersistence in various cases (Thomas, 2019), they are not suitable to study the Euler-Poincaré Surfaces. The reason for this is that matching and interleaving distances are based on $L_\infty$ metrics, and do not see the quantity of generators of small interleaving distances. However, Euler-Poincaré Surfaces by definition depends on the rank of the homology groups, and the quantity of the generators (Betti numbers) are crucial in their definition even if they exhibit a very short lifespan. In particular, $L_\infty$-based metrics find the maximum distance between the essential generators, but such metrics fail to capture the quantity of small generators. That is, while the interleaving or matching distance between the multipersistence modules of the bifiltrations $\{\widehat{\mathbf{C}}^+\}$ and $\{\widehat{\mathbf{C}}^-\}$ is small, there might be a very large distance between the induced Euler-Poincaré Surfaces $\mathbb{E}^+$ and $\mathbb{E}^-$. In turn, $L_p$-based distances are successful to capture the rank even if they are small (noise) generators, while interleaving and matching distances are not very sensitive to these. (See Thomas (2019, Section 4.2) for further discussion.) This is why we introduce the weak $L_1$ metric above which naturally suits to study the Euler-Poincaré Surfaces and similar vectorizations in the theory. Furthermore, such weak $L_1$ metric can be considered a straightforward generalization of $L_1$-based norm for multiparameter persistence.

**Remark C.2 (Implications of the Stability Result).** Note that our stability result (Theorem 3.1) directly implies that the distances between multiparameter PDs control the distance between the resulting Euler-Poincaré Surfaces. By combining with the stability result for PDs (Cohen-Steiner et al., 2007), one can conclude that the small changes in the MP filtering function $F : \mathcal{V}_t \mapsto \mathbb{R}^d$ or the small changes in the input data can result only in a small change in DEPS surfaces. In particular, by Cohen-Steiner et al. (2007), for single parameter persistence, we have

$$\mathcal{W}(D^f(\mathbf{C}), D^g(\mathbf{C})) \leq C\dot{d}(f, g),$$

where $D^f$ is the persistence diagram for filtering function $f$, and $d(f, g)$ is the distance between the functions $f$ and $g$. The way we define our weak $L_1$-metric enables us to use this single persistence result in this context as follows. The column distance $\mathfrak{D}_c$ is the sum of single PD for each column. Each column, (say $j^{th}$ column) is represented by a single variable filtering function $F(t, \beta_j)$. Then, the changes in $F(t, \beta_j)$ reflects in the column distance for PDs corresponding to $j^{th}$-column. This means if we have a small change in $F(t, \beta_j)$ for each $j$, then the column distance for MP will be small. So is the row distance induced by $F(\alpha_i, s)$.

Similarly, if we have two MP filtering functions $F^\pm : \mathcal{V} \to \mathbb{R}^2$, they will induce two filtrations $\mathbf{C}^\pm$. When $d(F^+, F^-)$ is small, then by Cohen-Steiner et al. (2007), $\mathfrak{D}(\widehat{\mathbf{C}}^+, \widehat{\mathbf{C}}^-)$ is small. When we apply our stability theorem to these filtrations, we see that the distance between the induced DEPS surfaces $\|\mathbb{E}^+ - \mathbb{E}^-\|_{1,1}$ will be small, too. Since small change in the data can be converted to small change in the filtering functions on the same data, the result can also be interpreted as the small change in the data results in small change in the induced DEPS surfaces.

**Remark C.3 (DEPS through the Probabilistic Perspective).** Furthermore, the new time-aware MP summary can also be assessed from a probabilistic perspective, if we view a multivariate time series $\mathcal{X}$ as real-valued random process indexed by time $t$ and defined on a probability space $(\Sigma, \mathcal{A}, \mathbb{P})$. For $1 \leq J \leq L \leq \infty$, define the $\sigma$-field $F_J^L \subset \mathcal{A}$ such that $F_J^L = \sigma(\boldsymbol{X}_k, J \leq k \leq L \ (k \in \mathbb{Z}))$, i.e. $F_J^L$ is a $\sigma$-field generated by $\boldsymbol{X}_J, \boldsymbol{X}_{J+1}, \ldots, \boldsymbol{X}_L$. Then, $\mathbb{E}^t$ is a real-valued $F_1^t$-measurable random matrix and DEPS $\{\mathbb{E}^t\}_{t=1}^T$ is $\mathbb{F}$-measurable random matrix, where $\mathbb{F} = (F_1^k)_{k>0}$ is a filtration of $\sigma$-fields indexed over time. Given the results of Thoppe et al. (2016) on weak convergence of the Euler characteristic $\chi(\mathbf{C})$ for dynamic Erdős-Rényi graphs to the Ornstein-Uhlenbeck process, this opens a potential path to establish asymptotic distribution of $\mathbb{E}^t$.

**Remark C.4 (DEPS for Higher Dimensions).** DEPS can be naturally expanded to a case when we consider filtrations along more than two geometric dimensions. In such a case, instead of a $m \times n$-matrix, DEPS will be a tensor. Our stability result (i.e., Theorem 3.1) verbatim extends to such a case.

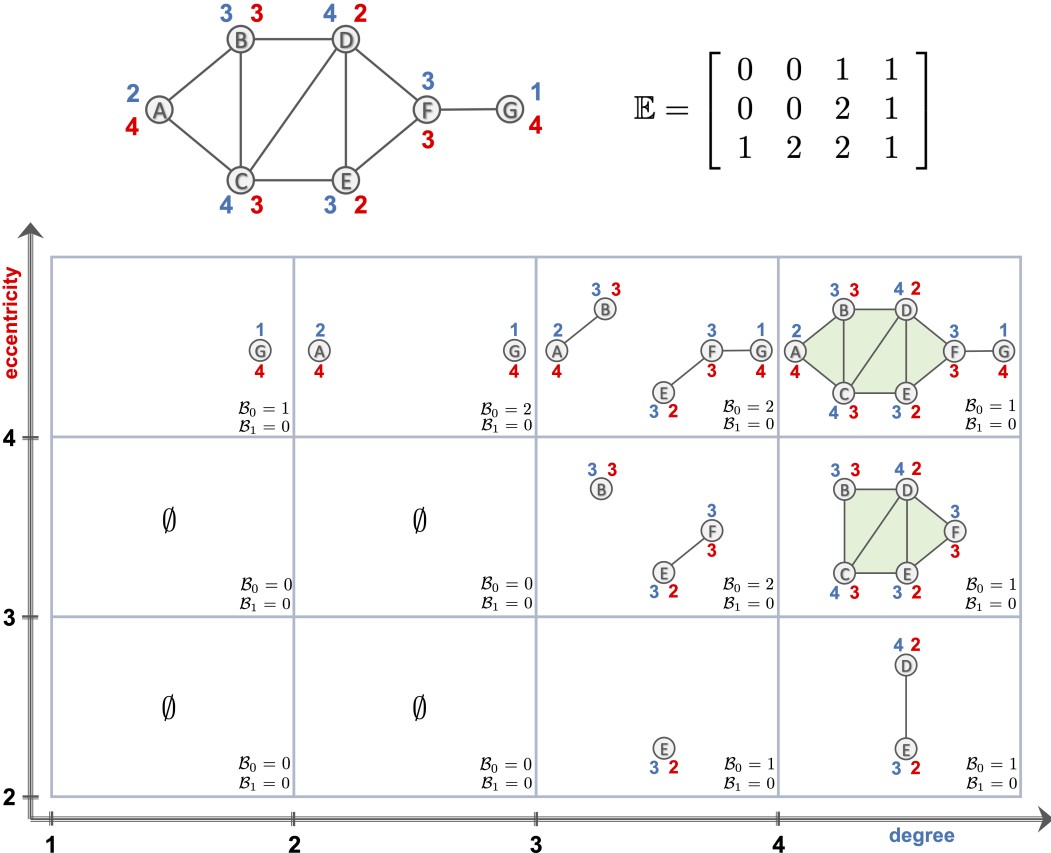

Figure 2: Illustration of multidimensional persistence (i.e., multipersistence). In the original graph (the top left panel), red numbers represent the eccentricity of the node, while blue numbers are the node degrees. The top right panel depicts the corresponding Euler-Poincaré Surface. Shape properties of the top graph are evaluated along two geometric dimensions (i.e., degree and eccentricity). In particular, for each row, the rightmost graph is filtered by degree function. For each column, the top graph is filtered by eccentricity function. Each cell includes Betti numbers $\mathcal{B}_0$ and $\mathcal{B}_1$. Since $\mathcal{B}_1$ and $\mathcal{B}_2$ are 0 for each cell, we get $\mathbb{E}_{ij} = \mathcal{B}_0$.

# D  REPRODUCIBILITY

## D.1  DATASETS

We use three different datasets, from transportation, Ethereum blockchain, and biosurveillance domains, to examine the performance of the proposed TAMP-S2GCNets methodology:

1. **Transportation:** The transportation and traffic data for the U.S. State of California, from the freeway Performance Measurement System (PeMS) data sources (i.e., PeMSD3, PeMSD4, PeMSD8) (Chen et al., 2001), naturally produces a dynamic network where each node is a loop detector and each edge represents a freeway between two nodes. We extract aggregated data to 5 minutes, thus generating dynamic networks with 26,208, 16,992 and 17,856 nets in PeMSD3, PeMSD4 and PeMSD8 datasets, respectively. To capture spatio-temporal dependencies, we rebuild the traffic graph structure, at time $t$, and compute edge weights $w_{t,uv}$, between pair nodes $(u, v)$, via the Radial Basis Function $w_{t,uv} = e^{-||x_{t,u} - x_{t,v}||^2/\gamma}$; where $\gamma = 1.0$ and each node $x_{t,\cdot}$ has features speed, occupancy and flow rate. To investigate the dynamic of the traffic graph topology, we restrict the final graphs to only keep edges whose are below or equal to threshold $\alpha = 0.01$.

2. **Blockchain:** Ethereum blockchain data contain token assets which naturally represents network layers of the Ethereum network (di Angelo & Salzer, 2020; Web, e), in which nodes and edges are addresses of users and digital transactions, respectively. Using the publicly available blockchain, we extract dynamic networks from three tokens with market value above $100,000,000: Bytom, Decentraland and Golem. In our experiments, we extract dynamic networks by daily transactions and use daily closed prices (Web, d), hence, creating a graph/net for each day. Since each token is created at different date, in all cases last day is 05/07/2018, the dynamic networks vary in the number of nets: Bytom (285 nets), Decentraland (206 nets) and Golem (443 nets). For the sake of reasonable computation time, we reduce the original net size using the maximum weight subgraph approximation method of Vassilevska et al. (2006) from more than 400,000 nodes, average of 120,000 edges, to 100 nodes using number of transactions and price-volume as edge weights.

3. **Biosurveillance:** SARS-CoV-2, which is the virus that causes COVID-19, has spread around every place in the world. To analyze the progression of coronavirus disease we create a dynamic network based on daily records on COVID-19 cases and hospitalizations from the CovidActNow project (Web, c) and Johns Hopkins University (Web, f; Dong et al., 2020), and population numbers from the U.S. Census Bureau (Web, b). We choose data at county level from California and Texas states as our cases of study, and focus on forecasting the number of hospitalized patients. Similarly to (i) we rebuild the coronavirus-spread graph structure, at day-time $t$, based on the county adjacency graph (Web, a) and Radial Basis Function between pair of nodes. We build a dynamic network for each state, each dynamic network contains 335 nets. The number of confirmed COVID-19 cases and county population size serves as node features, and final graphs only keep edges whose are below or equal to the threshold $\alpha = 0.01$.

More details on experimental settings and filtering functions are provided in Section D.2. Table 6 summarize the characteristics of all datasets used in our experiments.

Table 6: Summary of datasets used in time series forecasting task.

| Type | Dataset | No. Nodes | No. avg Edges | Time interval |
|------|---------|-----------|---------------|---------------|
| Traffic | PeMSD3 | 358 | 55.91 | 09/01/2018 - 11/30/2018 |
| Traffic | PeMSD4 | 307 | 26.32 | 01/01/2018 - 28/02/2018 |
| Traffic | PeMSD8 | 170 | 18.51 | 01/07/2016 - 31/08/2016 |
| Ethereum | Bytom | 100 | 9.98 | 27/07/2017 - 07/05/2018 |
| Ethereum | Decentraland | 100 | 16.94 | 14/10/2017 - 07/05/2018 |
| Ethereum | Golem | 100 | 20.58 | 18/02/2017 - 07/05/2018 |
| COVID-19 | CA | 55 | 56.15 | 01/02/2020 - 31/12/2020 |
| COVID-19 | TX | 251 | 929.80 | 01/02/2020 - 31/12/2020 |

## D.2 EXPERIMENTAL SETTINGS

We implement our TAMP-S2GCNets with Pytorch framework on NVIDIA GeForce RTX 3090 GPU. Further, for all datasets, TAMP-S2GCNets is trained end-to-end by using the Adam optimizer with a L1 loss function. The tuning of our proposed TAMP-S2GCNets on each dataset is done via grid search over a fixed set of choices and the same cross validation setup is used to tune baselines. We compare TAMP-S2GCNets with 13 types of state-of-the-art methods, including

FC-LSTM (Sutskever et al., 2014), SFM (Zhang et al., 2017), N-BEATS (Oreshkin et al., 2019), DCRNN (Li et al., 2018), LSTNet (Lai et al., 2018), STGCN (Yu et al., 2018), TCN (Bai et al., 2018), DeepState (Rangapuram et al., 2018), GraphWaveNet (Wu et al., 2019), DeepGLP (Sen et al., 2019), AGCRN (Bai et al., 2020), Z-GCNETs (Chen et al., 2021), and StemGNN (Cao et al., 2020).

**MODEL SPECIFICATIONS**  For **PeMSD3**, TAMP-S2GCNets consists of 2 layers whose hidden feature dimension is 64. The learning rate is $5e$-3 and the batch size is set as 64. The "cost" $d_{uu}^{t_a t_a}$ of staying in the same node $u$ (in the layer $t_a$) and the "cost" $d_{uu}^{t_a t_b}$ of jumping from the current node $u$ in layer $t_a$ to node $u$ in layer $t_b$ are set to be 0.1 and 0.2, respectively.

For **PeMSD4**, TAMP-S2GCNets consists of 2 layers whose hidden feature dimension is 128. The learning rate is $5e$-3 and the batch size is set as 64. The "cost" $d_{uu}^{t_a t_a}$ of staying in the same node $u$ (in the layer $t_a$) and the "cost" $d_{uu}^{t_a t_b}$ of jumping from the current node $u$ in layer $t_a$ to node $u$ in layer $t_b$ are set to be 0.1 and 0.2, respectively.

For **PeMSD8**, TAMP-S2GCNets consists of 2 layers whose hidden feature dimension is 64. The learning rate is $2e$-3 and the batch size is set to 16. The "cost" $d_{uu}^{t_a t_a}$ of staying in the same node $u$ (in the layer $t_a$) and the "cost" $d_{uu}^{t_a t_b}$ of jumping from the current node $u$ in layer $t_a$ to node $u$ in layer $t_b$ are set to be 0.1 and 0, respectively. The dimension of node embedding $d_c$ (in $\boldsymbol{E}_\phi$) are 10 (PeMSD3), 10 (PeMSD4), and 20 (PeMSD8), respectively. We set the dimension $K$ of Laplacian tensor $\tilde{\boldsymbol{S}}$ as 2 for all transportation networks.

For all **Ethereum token networks** (i.e., **Bytom**, **Decentraland**, and **Golem**), TAMP-S2GCNets consists of 2 layers whose hidden feature dimension is 32. The final hyperparameter setting is learning rate of $1e$-2 and batch size of 8. The "cost" $d_{uu}^{t_a t_a}$ of staying in the same node $u$ (in the layer $t_a$) and the "cost" $d_{uu}^{t_a t_b}$ of jumping from the current node $u$ in layer $t_a$ to node $u$ in layer $t_b$ are set to be 0.01 and 0.01, respectively. In addition, we set the embedding dimension $d_c$ in $\boldsymbol{E}_\phi$ as 1 for all Ethereum token networks. We set the dimension $K$ of Laplacian tensor $\tilde{\boldsymbol{S}}$ as 2, 3, 3 for Bytom, Decentraland, and Golem, respectively.

For **COVID-19 hospitalizations** in **CA** and **TX**, TAMP-S2GCNets consists of 2 layers whose hidden feature dimension is 128. The learning rate is set to $1e$-1 and $1e$-2 for CA and TX, respectively. The batch size is set to 8, and the embedding dimension $d_c$ in $\boldsymbol{E}_\phi$ is 10. The "cost" $d_{uu}^{t_a t_a}$ of staying in the same node $u$ (in the layer $t_a$) and the "cost" $d_{uu}^{t_a t_b}$ of jumping from the current node $u$ in layer $t_a$ to node $u$ in layer $t_b$ are set to be 0.1 and 0.2, respectively. We set the dimension $K$ of Laplacian tensor $\tilde{\boldsymbol{S}}$ as 2 for both CA and TX.

**ALTERNATIVE OPTIONS FOR DEPS WITHIN DEPSRL MODULE**  In our experiments, for all datasets, the CNN based model (i.e., $f_{\theta_j}$ where $j = \{1, 2\}$ in Time-Aware Multipersistence Euler-Poincaré Surface Representation Learning (DEPSRL) module) consists of 2 CNN layers. For transportation networks and COVID, The filter size, kernel size, and stride is set to be 8, 2, 2 respectively. For Ethereum token networks (i.e., Bytom, Decentraland, and Golem), the filter size, kernel size, and stride is set to be 8, 1, 2 respectively. We set the size of global average pooling and global max pooling as $3 \times 3$. Table 7 shows the optimal output dimensions of spatial graph convolutional layer ($Q_{\mathsf{Spa}}$), supragraph diffusion convolutional layer ($Q_{\mathsf{Sup}}$), and spatio-temporal feature transformation ($Q_{\mathsf{FT}}$) on all datasets.

Table 7: The output dimensions of spatial graph convolutional layer ($Q_{\mathsf{Spa}}$), supragraph diffusion convolutional layer ($Q_{\mathsf{Sup}}$), and spatio-temporal feature transformation ($Q_{\mathsf{FT}}$) on different datasets.

| Output dimension | PeMSD3 | PeMSD4 | PeMSD8 | Bytom | Decentraland | Golem | CA | TX |
|---|---|---|---|---|---|---|---|---|
| $Q_{\mathsf{Spa}}$ | $\frac{Q_{\mathsf{out}}}{4}$ | $\frac{Q_{\mathsf{out}}}{4}$ | $\frac{3Q_{\mathsf{out}}}{4}$ | $\frac{Q_{\mathsf{out}}}{2}$ | $\frac{Q_{\mathsf{out}}}{2}$ | $\frac{3Q_{\mathsf{out}}}{4}$ | $\frac{Q_{\mathsf{out}}}{4}$ | $\frac{Q_{\mathsf{out}}}{4}$ |
| $Q_{\mathsf{Sup}}$ | $\frac{Q_{\mathsf{out}}}{4}$ | $\frac{Q_{\mathsf{out}}}{4}$ | $\frac{Q_{\mathsf{out}}}{8}$ | $\frac{Q_{\mathsf{out}}}{4}$ | $\frac{Q_{\mathsf{out}}}{4}$ | $\frac{Q_{\mathsf{out}}}{8}$ | $\frac{Q_{\mathsf{out}}}{4}$ | $\frac{Q_{\mathsf{out}}}{4}$ |
| $Q_{\mathsf{FT}}$ | $\frac{Q_{\mathsf{out}}}{2}$ | $\frac{Q_{\mathsf{out}}}{2}$ | $\frac{Q_{\mathsf{out}}}{8}$ | $\frac{Q_{\mathsf{out}}}{4}$ | $\frac{Q_{\mathsf{out}}}{4}$ | $\frac{Q_{\mathsf{out}}}{8}$ | $\frac{Q_{\mathsf{out}}}{2}$ | $\frac{Q_{\mathsf{out}}}{2}$ |

Furthermore, in addition to the Time-Aware Multipersistence Euler-Poincaré Surface Representation Learning (DEPSRL) module (see Section 4.3 in our main paper):

$$\mathbf{\Phi}_{i,TAMP}^{(\ell+1)} = \oplus(f_{\text{GAP}}(f_{\theta_1}(\{\mathbb{E}^i\}_{i=1}^{\mathcal{T}})), f_{\text{GMP}}(f_{\theta_2}(\{\mathbb{E}^i\}_{i=1}^{\mathcal{T}}))),$$

we also have considered passing the aggregated DEPS as input to DEPSRL, that is, $f_{\theta_j}(\bar{\mathbb{E}})$, $j = 1, 2$, where $\bar{\mathbb{E}} = 1/\mathcal{T} \sum_{i=1}^{\mathcal{T}} \in \mathbb{R}^{m \times n} \mathbb{E}^i$.

As expected, DEPS $\{\mathbb{E}^i\}_{i=1}^{\mathcal{T}}$ tends to yield better performance within the DEPSRL module than its aggregated counterpart $\bar{\mathbb{E}}$, which can be explained by lower loss of time-conditioned topological information within DEPS. However, in the case of PeMSD4 dataset, we find that $\bar{\mathbb{E}}$ slightly improves the performance of TAMP-S2GCNets (i.e., the MAE of TAMP-S2GCNets based on the aggregated DEPS is 17.58 and the MAE of TAMP-S2GCNets based on DEPS is 17.69), which is likely due to higher data heterogeneity of PeMSD4 vs. other transportation datasets. In all other datasets, we report results based on DEPS.

**COMPARISON TO OTHER MP SUMMARIES** For fair comparison, computation of all MP representations (as shown in Table 11 and Table 4 (**right panel**) in our main paper) have been run on an 8-cores DO droplet machine with Intel Xeon Scalable processors at base frequency of 2.5 Ghz. Our source codes are available for revision[1]. Source codes for computation of MP-I and MP-L are freely available in Github[2]. We follow the filtering function recommendations for graphs, on both MP-I and MP-L, from supplementary material of Carrière & Blumberg (2020). Further technical details on the 8-core machine are publicly available[3].

**ON TYPES OF MULTIFILTRATIONS** Good filtrations should have rich range to get a fine resolution, and it should reflect an important property relevant to the question at hand. For multipersistence, for a good combination of two such filtering functions, one should choose functions uncorrelated. One approach is to choose one function from the domain of the question, and the other from strictly graph properties (e.g. degree, betweenness, etc.). We assess shape characteristics of $\mathcal{G}_t$ via filtering functions based on node degree (`Deg`), betweenness centrality (`Btwns`), and edge weights (`Power`). Edge weights for each dataset are computed as described in D.1; i.e using transactions between addresses (`Trns`), volume of transactions (`Vol`), traffic measurements (`Traffic`) and COVID-19 incidence (`Incidence`). Our analysis focuses on multivariate filtering function $F : \mathcal{V}_t \mapsto \mathbb{R}^d$ with $d = 2$. For `Deg` and `Btwns` the set of nondecreasing thresholds runs on values over nodes, whilst for filtering via power filtration (`Power`) of $\mathcal{G}_t$ nondecreasing thresholds runs on weighted paths between nodes. Notice that each pair-combination of filtering functions produces a specific MP representation in a form of the Euler-Poincaré surface, which impacts the algorithm performance, as shown in our multiple experiments. Table 12 presents the optimal multifiltration for each dataset.

# E    ADDITIONAL EXPERIMENTS AND RUNNING TIME

Table 8: Forecasting performance and standard deviations on PeMSD3, PeMSD4 and PeMSD8 datasets. All models are re-run using the original authors' code.

| Model | PeMSD3 | | | PeMSD4 | | | PeMSD8 | | |
|---|---|---|---|---|---|---|---|---|---|
| | MAE | RMSE | MAPE (%) | MAE | RMSE | MAPE (%) | MAE | RMSE | MAPE (%) |
| DCRNN (Li et al., 2018) | 18.18±0.15 | 30.31±0.25 | 18.91±0.82 | 24.70±0.22 | 38.12±0.26 | 17.12±0.37 | 17.86±0.03 | 27.83±0.05 | 11.45±0.03 |
| STGCN (Yu et al., 2018) | 17.49±0.46 | 30.12±0.70 | 17.15±0.45 | 22.70±0.64 | 35.50±0.75 | 14.59±0.21 | 18.02±0.14 | 27.83±0.20 | 11.40±0.10 |
| DeepState (Rangapuram et al., 2018) | 15.59±0.43 | **20.21**±0.60 | 18.69±0.22 | 26.50±0.25 | 33.00±0.67 | 15.40±0.23 | 19.34±0.11 | 27.18±0.22 | 16.00±0.10 |
| GraphWaveNet (Wu et al., 2019) | 19.85±0.03 | 32.94±0.18 | 19.31±0.49 | 26.85±0.03 | 39.70±0.04 | 17.29±0.24 | 19.13±0.08 | 28.16±0.07 | 12.68±0.57 |
| AGCRN (Bai et al., 2020) | 14.22±0.31 | 24.03±0.36 | 13.89±0.29 | 17.78±0.15 | 29.17±0.09 | 11.79±0.11 | 14.59±0.40 | 23.06±0.33 | 9.29±0.11 |
| Z-GCNETs (Chen et al., 2021) | 14.20±0.33 | 25.29±0.53 | 13.88±0.23 | 18.05±0.20 | 29.08±0.19 | 11.79±0.08 | 14.52±0.15 | 23.00±0.20 | 9.28±0.31 |
| **TAMP-S2GCNets (us)** | **13.91**±0.16 | 23.77±0.32 | **13.40**±0.39 | 17.58±0.20 | **28.56**±0.28 | **11.01**±0.10 | **13.77**±0.08 | **21.70**±0.25 | **8.99**±0.15 |

**Why MP Helps? More Experiments** To validate gains (if any) delivered by multifiltration compared to the single filtration persistence and ensemble of multiple stacked together single parameter

---

Table 9: The TAMP-S2GCNets ablation study on Golem (MAPE) and CA. Here *, **, *** denote significant, statistically significant, highly statistically significant results.

| Architecture | Dataset | |
|---|---|---|
| | Golem (MAPE) | CA (RMSE) |
| **TAMP-S2GCNets** | 20.10% | 371.60 |
| W/o DEPSRL | 20.37% | *380.18 |
| W/o GCN$_{Spa}$ | **21.51% | *378.87 |
| W/o GCN$_{Sup}$ | *20.72% | **382.63 |
| W/o FT | **21.08% | ***383.37 |

filtrations, we also report the results of single filtration, filtration ensemble, and multifiltration on Decentraland (see Table 10). (In case of a filtration ensemble, we aggregate two single filtrations and feed the ensemble representation into the model.) We find that MP substantially outperforms both the single filtration persistence and the filtration ensemble. In addition, the MP results are generally more stable than ones based on the single filtration and filtration ensemble. This result can be expected, as by construction of DEPS, each individual single parameter persistence corresponds either to a row or a column of DEPS, that is, we have only a one-dimensional fingerprint of the data. In contrast, in MP and its DEPS summary, we obtain a two-dimensional fingerprint of the data, meaning any cell in the output simultaneously related to the all 8 neighboring cells (East, West, Northeast, etc.). As such, MP provides a 2D resolution of the data space and delivers much finer information on the underlyng data topology.

Table 10: MAPE (in%) (standard deviation) of TAMP-S2GCNets with single filtration, filtration ensemble, and multifiltration persistence.

| TAMP-S2GCNets on Decentraland | | |
|---|---|---|
| **Single filtration** | **Filtration ensemble** | **Multifiltration** |
| - | [Deg, Btwns]: 24.35±1.78 | Deg & Btwns: 21.13±1.57 |
| Deg: 25.18±2.00 | [Btwns, Power$_{Trns}$]: 23.00±2.13 | Btwns & Power$_{Trns}$: 20.00±1.51 |
| Btwns: 24.93±1.88 | [Btwns, Power$_{Vol}$]: 22.70±2.90 | Btwns & Power$_{Vol}$: 21.59±1.76 |
| Power$_{Trns}$: 23.81±1.68 | [Deg, Power$_{Trns}$]: 21.81±1.65 | Deg & Power$_{Trns}$: **19.89±1.69** |
| Power$_{Vol}$: 22.15±2.70 | [Deg, Power$_{Vol}$]: 21.58±3.03 | Deg & Power$_{Vol}$: 20.83±1.86 |

Table 11 shows the computational costs and performance of different MP summaries on Decentraland and Golem datasets. We observe that our new time-aware MP summary DEPS not only outperforms the existing MP summaries (i.e., MP-I and MP-L) but also induces substantially lower computational costs. These findings demonstrate that our DEPS can effectively capture salient time-aware topological information with lower time and memory cost.

Table 11: MAPE (%) and computational time (in second) on Decentraland and Golem for TAMP-S2GCNets with our DEPS and MP-I of Carrière & Blumberg (2020) and MP-L of Vipond (2020), as MP summaries.

| MP summary | Decentraland | Golem |
|---|---|---|
| MP-I (Carrière & Blumberg, 2020) | 22.58 (366.71s) | 22.00 (850.20s) |
| MP-L (Vipond, 2020) | 23.10 (489.50s) | 22.05 (1137.09s) |
| DEPS$_{Deg\ \&\ Btwns}$ | 21.13 (43.25s) | 21.05 (126.15s) |
| DEPS$_{Deg\ \&\ Power_{Trns}}$ | **19.89** (36.90s) | **20.10** (112.61s) |
| DEPS$_{Btwns\ \&\ Power_{Trns}}$ | 20.00 (**25.34**s) | 21.00 (**85.87**s) |

**COVID-19 Biosurveillance: More Experiments** Finally, Figure 3 depicts the results on 15-day ahead forecasting of hospitalizations in California on a county level basis, using the top 3 models (i.e., AGCRN (Bai et al., 2020), StemGNN (Cao et al., 2020), and our TAMP-S2GCNets). (Here, to differentiate among RMSEs on a county level basis, we omit the map for DCRNN (Li et al.,

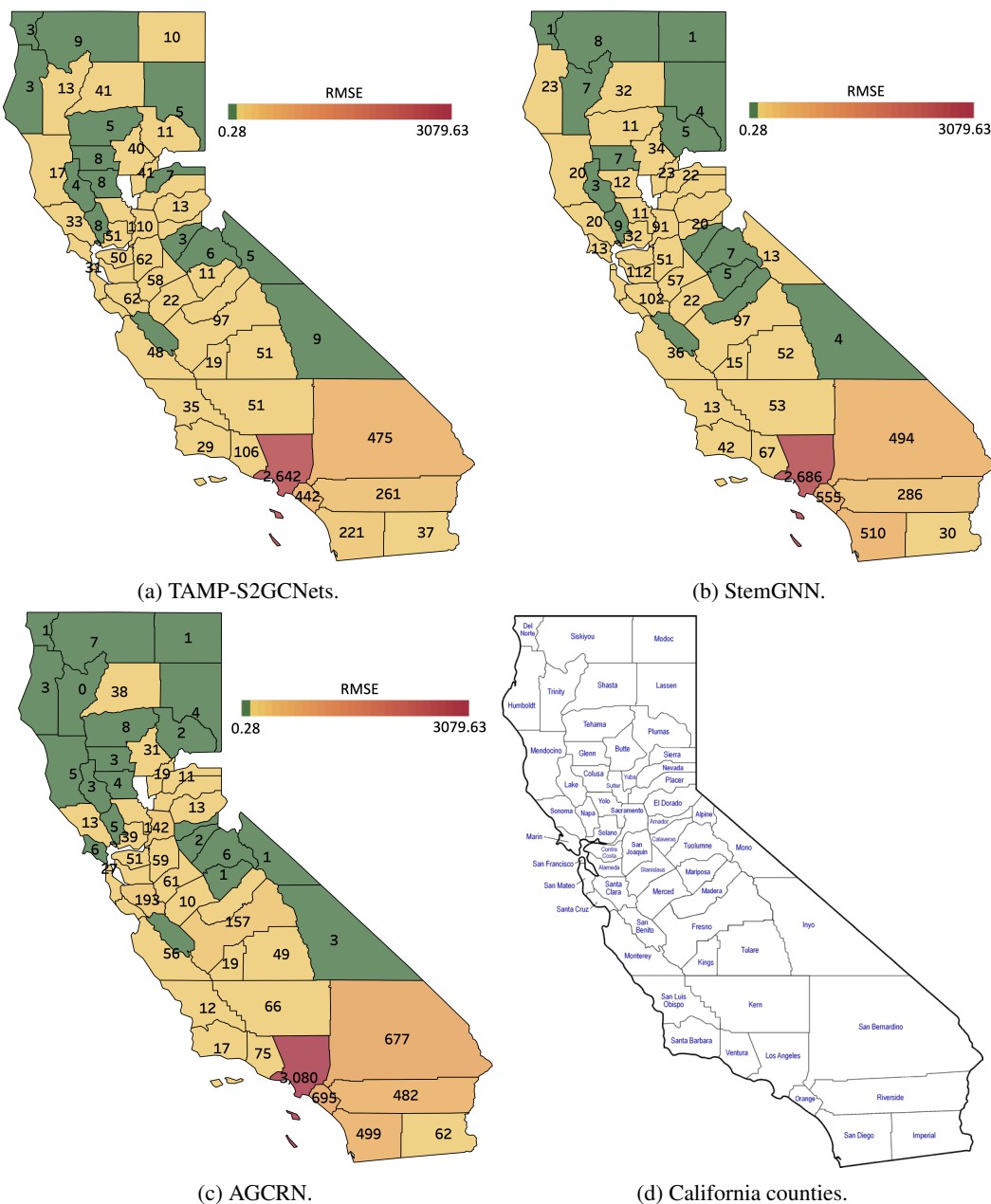

Figure 3: Maps of RMSE values for 15-day ahead forecasts of COVID-19 hospitalizations on a county level in California and counties in California. (a) TAMP-S2GCNets. (b) StemGNN. (c) AGCRN. (d) California counties (from California Department of Education).

2018) due to its lower predictive performance, relative to RMSE scale of AGCRN, StemGNN, and our TAMP-S2GCNets.) Overall, TAMP-S2GCNets and StemGNN tend to perform similarly for less populated counties in Northern California, with TAMP-S2GCNets being slightly better than StemGNN over the Pacific Coast and StemGNN yeilding some edge in the North East. In turn, AGCRN tends to deliver more competitive performance in North-Eastern California, except of Shasta county.

However, performance among models starts to differ drastically in more populated counties and, in particular, in Southern California where it is much harder to forecast COVID-19 hospitalizations due to higher population density and more disparate socio-economic factors. Here, TAMP-S2GCNets

Table 12: Types of multifiltration for all datasets.

| Dataset | Multifiltration |
|---------|-----------------|
| PeMSD3 | `Deg` & `Btwns` |
| PeMSD4 | `Deg` & `Btwns` |
| PeMSD8 | `Btwns` & `Power`$_{\text{Traffic}}$ |
| Bytom | `Btwns` & `Power`$_{\text{Trns}}$ |
| Decentraland | `Deg` & `Power`$_{\text{Trns}}$ |
| Golem | `Deg` & `Power`$_{\text{Trns}}$ |
| COVID-19 CA | `Btwns` & `Power`$_{\text{Incidence}}$ |
| COVID-19 TX | `Deg` & `Power`$_{\text{Incidence}}$ |

substantially outperforms both StemGNN and AGCRN almost in all counties of Southern California, yielding predictive gains from 3.8% vs. StemGNN and 29.8% vs. AGCRN in San Bernardino to 57.7% vs. StemGNN and 55.7% vs. AGCRN in San Diego. While all models perform very poorly in Los Angeles county (i.e., one of the most populated counties), TAMP-S2GCNets still yields the lowest RMSE. One important phenomenon we observe is that TAMP-S2GCNets systematically tends to substantially outperform StemGNN and AGCRN in more highly populated counties, including both Southern and Northern California. For instance, the predictive gains of TAMP-S2GCNets in Santa Clara are 39.2% vs. StemGNN and 67.9% vs. AGCRN. These findings indicate that time-aware MP topological information contained in TAMP-S2GCNets can address hidden relationships among spatio-temporal factors contributing to COVID-19 dynamics which other models cannot.

## F  ADDITIONAL DETAILS OF TAMP-S2GCNETS ARCHITECTURE AND EXPERIMENTS

**Remark F.1 (Spatio-Temporal Feature Transformation).** We utilize and share the learnable node embedding $\boldsymbol{E}_\phi$ to (i) construct the normalized self-adaptive adjacency matrix $\boldsymbol{S}$ and (ii) project the target node from the input feature space (in spatial and temporal dimensions) into a high-level latent space (together with $\boldsymbol{\Theta}_{FT}$). Hence, this shared-update weights can not only mitigate the risk of overfitting but allow us to capture shared patterns among spatial-temporal features and graph structures over the time axis. The learnable node embedding $\boldsymbol{E}_\phi$ in TAMP-S2GCNets can be obtained from the backward propagation through time.

We use Einstein summation convention in Eq. 1 for the multiplication of $X^{\mathcal{T}} \in \mathbb{R}^{b \times \mathcal{T} \times N \times P}$ (where $b$ is the batch size) and $\boldsymbol{E}_\phi \boldsymbol{\Theta}_{FT}^{(\ell)}$. The implementation of matrix multiplication in Eq. 1 is as follows: (i) we first perform the matrix multiplication between $\boldsymbol{E}_\phi \in \mathbb{R}^{N \times d_c}$ and $\boldsymbol{\Theta}_{FT}^{(\ell)} \in \mathbb{R}^{d_c \times P \times Q_{\text{FT}}}$ to get an intermediate weight term denoted by $\tilde{\boldsymbol{\Theta}}_{FT}^{(\ell)} \in \mathbb{R}^{N \times P \times Q_{\text{FT}}}$; (ii) then we compute the matrix multiplication between $\mathbf{X}^{\mathcal{T}}$ and $\tilde{\boldsymbol{\Theta}}_{FT}^{(\ell)}$ through Einstein summation convention, i.e., in Pytorch, output = torch.einsum('btni, nio $\rightarrow$ btno', X, intermediate_weight_term) (where X is $\mathbf{X}^{\mathcal{T}}$ and intermediate_weight_term is $\tilde{\boldsymbol{\Theta}}_{FT}^{(\ell)}$).

**Remark F.2 (Normalized Self-Adaptive Adjacency Matrix).** We utilize the node embedding dictionaries $\boldsymbol{E}_\phi \in \mathbb{R}^{N \times d_c}$ to construct the normalized self-adaptive adjacency matrix $\boldsymbol{S}$ via multiplying $\boldsymbol{E}_\phi$ and $\boldsymbol{E}_\phi^\top$. As such, construction of the normalized self-adaptive adjacency matrix $\boldsymbol{S}$ does not impose any requirements on density of the input graph. That is, our graph learning architecture (i.e., the self-adaptive adjacency matrix construction mechanism) overcomes this problem of the dense input graph met by GAT.

To explore more computationally efficient mechanisms of learning the normalized self-adaptive adjacency matrix, in the future work, we will (i) apply the sparse sampling strategies to the graph learning architecture; (ii) learn the graph based on the original graph structure information, i.e., we only infer the dependencies between the adjacent nodes (i.e., there exists an edge between them in the original graph) via multiplying $e_{u,\phi}$ and $e_{v,\phi}^\top$; (iii) construct hypergraph representation instead of graph which not only alleviate the computational complexity but capture heterogeneous higher-order structures information.

For $\tilde{\boldsymbol{S}}$, in our experiments, we performed an extensive grid search for the hyperparameter $K$ (i.e., truncation order). We find the optimal $K$ of 2 for the transportation network, the optimal $K$ of 2 for COVID-19 network dataset, and the optimal $K$ of 3 for Ethereum blockchain network. That is, for larger networks the truncation order $K$ tends to be lower so that the computational costs are mitigated.

Furthermore, we have performed ablation studies on PeMSD4 and PeMSD8 to assess utility of the normalized self-adaptive adjacency matrix $\boldsymbol{S}$ in TAMP-S2GCNets. The results in Tables 13 and 14 indicate that TAMP-S2GCNets outperforms TAMP-S2GCNets with the plain adjacency matrix (instead of the normalized self-adaptive adjacency matrix) for both PeMSD4 and PeMSD8 across all the evaluation metrics. The results illustrate the importance of learning the node-specific patterns/embeddings over the spatial-temporal time series dataset.

Table 13: Ablation study of the normalized self-adaptive adjacency matrix on PeMSD4.

| Architecture | MAE | RMSE | MAPE (%) |
|---|---|---|---|
| TAMP-S2GCNets | 17.58 | 28.56 | 11.01 |
| TAMP-S2GCNets with plain adjacency matrix | 17.69 | 28.66 | 11.74 |

Table 14: Ablation study of the normalized self-adaptive adjacency matrix on PeMSD8.

| Architecture | MAE | RMSE | MAPE (%) |
|---|---|---|---|
| TAMP-S2GCNets | 13.77 | 21.70 | 8.99 |
| TAMP-S2GCNets with plain adjacency matrix | 13.94 | 21.98 | 9.09 |

**Remark F.3 (Global Average Pooling and Global Max Pooling).** We utilize the global average pooling (GAP) to preserve information about the Euler-Poincaré surfaces in a fixed-size representation through averaging of the learnt topological embeddings. In our experiments, we find that adding global max pooling (GMP) tends to strengthen the Euler-Poincaré surfaces representation and to improve the overall performance. Before GAP and GMP, we first use CNN base models (i.e., $f_{\theta_1}(\cdot)$ and $f_{\theta_2}(\cdot)$) to learn the topological features of Euler-Poincaré surfaces (i.e., $f_{\theta_i}(\{\mathbb{E}^i\}_{i=1}^{\mathcal{T}})$, where $i = \{1, 2\}$); then we employ GAP and GMP to the outputs of CNN base models, respectively. We have conducted the ablation study to evaluate the performance of TAMP-S2GCNets without (w/o) GAP or GMP on Golem. From Table 15, we find that applying both GAP and GMP enhances prediction performance.

Table 15: Ablation study on global average pooling and global max pooling. Here $^*$ denotes significant results.

| Architecture | TAMP-S2GCNets | TAMP-S2GCNets w/o GAP | TAMP-S2GCNets w/o GMP |
|---|---|---|---|
| MAPE (%) | *20.10 | 20.83 | 20.57 |

**Remark F.4 (Supra-Laplacian).** In our experiments, for large-scale networks, we apply sparse sampling strategy for the supra-Laplacian construction. For example, on transportation network such as PeMSD3, we randomly select $\tau = 3$ timestamps out of the $\mathcal{T} = 12$ timestamps in the sliding window; as a result, we feed the sparse supra-Laplacian (i.e., $3N \times 3N$; where $N$ is the number of nodes) into the supragraph diffusion convolutional layer. For smaller networks such as Ethereum token networks, we utilize the $\mathcal{T}$ timestamps (i.e., the whole sliding window information) to construct the supra-Laplacian. In the future we will explore various graph sparsification techniques, e.g., subsampling, to reduce computational costs for larger networks.

**Remark F.5 (More Details on the Framework of TAMP-S2GCNets in Figure 1).** In Figure 1, the input is the target network over past 3 time slices ($\{\mathcal{G}_{t-2}, \mathcal{G}_{t-1}, \mathcal{G}_t\}$) and our TAMP-S2GCNets consists of 5 components. That is, (i) equipped with node feature matrix $X_t$, we apply spatial graph convolutional layer (i.e., GCN Layer in Figure 1) on $\mathcal{G}_t$ extracts spatial information at time $t$; (ii) feature transformation (i.e., FT in Figure 1) learns representation of the spatio-temporal data $\mathbf{X}^{\mathcal{T}}$ over a sliding window of size $\mathcal{T}$ (in Figure 1, for better understanding the operation procedure, we

set $\mathcal{T} = 3$, i.e., $\mathbf{X}^{\mathcal{T}} = \{X_{t-2}, X_{t-1}, X_t\}$); (iii) we create a multiplex network based on the input networks (i.e., $\{\mathcal{G}_{t-2}, \mathcal{G}_{t-1}, \mathcal{G}_t\}$) and construct the corresponding supra-Laplacian; lastly, we feed the supra-Laplacian into supragraph convolutional module (i.e., Supra GCN Layer in Figure 1); (iv) in Figure 1, we consider two types of Euler-Poincaré Surface Representation (DEPS), i.e., $\text{DEPS}_{f,g}$ and $\text{DEPS}_{f,h}$ are Euler-Poincaré surface representations under different multifiltrations, where $f, h, g$ represent different types of filtrations; in (iv), we feed Euler-Poincaré surface representations into CNN base models and employ global average/max pooling to the outputs of CNN base models. Finally, we combine these embeddings to obtain the final embedding, where is fed into GRU module for multi-step forecasting.

## G    ADDITIONAL COMPUTATIONAL COMPLEXITY AND TRAINING TIME COMPARISON

**Remark G.1 (Training Time Comparison).**  We report the average training time (per epoch in seconds) for our TAMP-S2GCNets and 4 baselines on PeMSD4 and Decentraland datasets in Tables 16 and 17. In terms of training time, TAMP-S2GCNets runs slightly slower than the baselines but the demonstrated running time is comparable to the baselines. Although the time per epoch is slightly higher for TAMP-S2GCNets, we find that the performance of TAMP-S2GCNets is significantly better than the state-of-the-art baselines. As such, this extra time appears to be worth it.

Table 16: Average training time (per epoch in seconds) for our TAMP-S2GCNets and 4 baselines on PeMSD4 dataset, where ** denotes statistically significant results.

|  | **TAMP-S2GCNets** | **StemGNN** | **Z-GCNets** | **AGCRN** | **DCRNN** |
|---|---|---|---|---|---|
| Average time | 40.30 s | 30.12 s | 37.53 s | 28.05 s | 28.79 s |
| RMSE | **28.56 | 31.83 | 29.08 | 29.17 | 38.12 |

Table 17: Average training time (per epoch in seconds) for our TAMP-S2GCNets and 4 baselines on Decentraland dataset, where *** denotes highly statistically significant results.

|  | **TAMP-S2GCNets** | **StemGNN** | **Z-GCNets** | **AGCRN** | **DCRNN** |
|---|---|---|---|---|---|
| Average time | 3.10 s | 2.55 s | 2.09 s | 2.03 s | 2.21 s |
| MAPE (%) | ***19.89 | 28.37 | 23.81 | 26.75 | 27.69 |

**Remark G.2 (More Details on Computational Complexity of Topological Summaries).**  Computational complexity of Dynamic Euler-Poincaré Surfaces (DEPS) is much lower than that of persistence diagrams. This is because Euler Characteristics is an alternating sum of the Betti numbers, and one does not need to compute the persistence diagrams to compute the Betti numbers. The computational complexity of a single persistence diagram $PD_k$ is $\mathcal{O}(\mathcal{N}^3)$, where $\mathcal{N}$ is the number of $k$-simplices (Otter et al., 2017). On the other hand, by using sparse matrix methods, computational complexity for the Euler Characteristics of a simplicial complex with $\mathcal{M}$ simplices is $\mathcal{O}(\mathcal{M})$ (Edelsbrunner & Parsa, 2014). If $p$ is the resolution size of the multipersistence grid, the resulting complexity for DEPS is $\mathcal{O}(p^2\mathcal{M})$. However, since one does not need to compute the persistence diagrams to obtain the Betti numbers, faster algorithms are possible. For instance, Lesnick & Wright (2019) developed a much faster algorithm (RIVET software) to compute Betti numbers in multipersistence setting. This reduction approach leads to computational complexity of $\mathcal{O}(R^3)$, where $R$ is the size of the filtered complex giving the minimal representation for the bipersistence module and $R$ is far less than the original number of simplices. Furthermore, by utilizing sparse matrix reductions, recent results of Kerber & Rolle (2021) significantly improved the RIVET approach. In particular, computational complexity for multipersistence landscapes with RIVET would be $\mathcal{O}(R^5)$, where $R$ is again the size of the reduced filtered complex (Vipond, 2020).

We have provided both computational complexity and running time of 8 topological summaries on Decentraland and Golem (see Table 18).

The computational complexity of the overall approach is: $\mathcal{O}(N^2 + N\mathcal{T}F_N Q_{\text{sup}} + N\mathcal{T}^2 Q_{\text{sup}}/2 + Q_{\text{sup}} \sum_{\ell=t-\mathcal{T}}^{t-1} M^{(\ell)} + R^3 + W_{GRU})$. That is, we have: (i) spatial graph convolution module: $\mathcal{O}(N^2)$

Table 18: Computational complexity and running time (in seconds) comparison on Decentraland dataset.

| Persistence summary[a] | Running time | MAPE | Computational complexity |
|---|---|---|---|
| MP-I (Carrière & Blumberg, 2020) | 366.71 s | 22.58 | $\mathcal{O}(p^2QB)$ [b] |
| MP-L (Vipond, 2020) | 489.50 s | 23.10 | $\mathcal{O}(R^5)$ [c] |
| DEPS$_{\text{Deg \& Btwns}}$ | 43.25 s | 21.13 | $\mathcal{O}(p^2\mathcal{M})$ |
| DEPS$_{\text{Deg \& Power}_{\text{Tms}}}$ | 36.90 s | 19.89 | or |
| DEPS$_{\text{Btwns \& Power}_{\text{Tms}}}$ | 25.34 s | 20.00 | $\mathcal{O}(R^3)$ [d] |
| PI$_{\text{Deg}}$ (Adams et al., 2017) | 11.43 s | 23.81 | |
| PI$_{\text{Btwns}}$ (Adams et al., 2017) | 10.01 s | 24.93 | $\mathcal{O}(p^2)$ [e] |
| PI$_{\text{Power}_{\text{Tms}}}$ (Adams et al., 2017) | 12.79 s | 25.18 | |

[a]Deg: node degree; Btwns: node betweenness centrality; Power: edge weight; Trns: transaction between addresses. Please refer to *on types of filtrations* (Appendix D.2) for a more detailed description of types of multifiltration.

[b]Where $p$ is the resolution of MP-I, $Q$ is the number of lines, $B$ is the maximum number of bars of the fibered barcodes.

[c]Where $R$ is the size of the reduced filtered complex (see previous paragraph). $\mathcal{M}$ is the number of simplices in the simplicial complex.

[d]The reported time for DEPS is given with the $\mathcal{O}(p^2 \times \mathcal{M})$ algorithm.

[e]This computational complexity is for the time after the computation of persistence diagrams.

(where $N$ is the number of nodes); (ii) supragraph diffusion convolution module: $\mathcal{O}(N\mathcal{T}F_N Q_{\text{sup}} + N\mathcal{T}^2 Q_{\text{sup}}/2 + Q_{\text{sup}} \sum_{\ell=t-\mathcal{T}}^{t-1} M^{(\ell)})$ (where $F_N$ is the number of node attribute features, $\mathcal{T}$ denotes the sliding window size, $Q_{\text{sup}}$ is the output dimension of the supragraph diffusion convolutional layer, and $M^{(\ell)}$ is the number of edges at the $\ell$-th layer); (iii) DEPS: $\mathcal{O}(R^3)$, where $R$ is the size of the reduced filtered complex; (iv) GRU: $\mathcal{O}(W_{GRU})$, where $W_{GRU}$ is the total number of parameters in the GRU.

