# OpenReview forum: "TAMP-S2GCNets: Coupling Time-Aware Multipersistence Knowledge Representation with Spatio-Supra Graph Convolutional Networks for Time-Series Forecasting"
_ICLR.cc/2022/Conference — ICLR 2022 Spotlight_

### Official Review · Reviewer_rDH8 · 2021-11-01

**Correctness:** 4
**Technical Novelty And Significance:** 3
**Empirical Novelty And Significance:** 4
**Recommendation:** 8
**Confidence:** 3

**Main Review:**

Strengths:
- This paper addresses a very relevant problem and one of broad impact on the community.
- Although at the very core, this paper is an attempt towards bringing together two emerging research directions, that combination is novel and unique. Furthermore, methodologically, there is a good degree of novelty in combining these two research directions (e.g., by introducing a new time-aware multi-parameter persistence invariant). Also, the authors make it clear how this work differs from previous contributions.
- The paper is technically sound.
- This is a well-written paper and generally well structured, making good use of appendices.
- The empirical evaluation is extensive and covers 3 different application domains. The empirical results are also very promising and supportive of the claimed advantages of the proposed approach.

Weaknesses:
- A discussion of the computational complexity of the proposed approach with respect to other time-series forecasting baselines is missing. The authors provide runtimes for the different multi-parameter persistence summary approaches but, a discussion of the computational complexity of the overall proposed approach would also be important.
- The supra-Laplacian is an NT x NT matrix, which raises questions about scalability that are not discussed in the paper.
- Some statements are not adequately supported by empirical evidence. For example: "The developed TAMP-S2GCNets model is shown to yield highly competitive forecasting performance on a wide range of datasets, with much lower computational costs." - this may be true across multi-parameter persistence summaries, but there are no empirical results to support such a claim across baselines.
- There are some choices that seem arbitrary or poorly justified. Perhaps the most obvious is in Eq. 6. Why are global average pooling and global max pooling concatenated together? What are both types of pooling needed?
- The text in Figure 1 is nearly impossible to read.

**Summary Of The Paper:**

This paper proposes the use of multi-parameter persistence (an emergent research topic in topological data analysis) to capture latent time-conditioned relations among nodes in a GNN. To do so, the authors introduce a dynamic Euler-Poincaré surface as a new multi-parameter persistence summary, and prove its stability and empirically show its computational efficiency. A "supra" graph convolution module is then proposed in order to allow for simultaneously learning co-evolving spatial and temporal correlations in the complex multivariate time-series data. The superior predictive performance of the proposed approach is shown empirically based on 3 different multi-variate time-series datasets from 3 different domains: highway traffic flow, Ethereum token prices, and COVID-19 hospitalizations.

**Summary Of The Review:**

Overall, this paper takes the important first steps towards a very promising research direction of bringing novel ideas from topological data analysis into GNN approaches to multi-variate time-series forecasting, and therefore could inspire and spawn several follow-up works. The paper is generally well-written. The experiments are extensive, cover 3 different application domains, and clearly demonstrate the potential of the proposed approach in particular, and this research direction in general. I have a few concerns regarding e.g. computational complexity that I would like to see more discussion on, but those do not prevent me from giving this paper a recommendation for acceptance.

---

### Official Review · Reviewer_8UH6 · 2021-11-03

**Correctness:** 3
**Technical Novelty And Significance:** 4
**Empirical Novelty And Significance:** 4
**Recommendation:** 8
**Confidence:** 4

**Main Review:**

The paper bridges time-aware deep learning with time-conditioned multiparameter persistence and this leads to the development of a new model TAMP-S2GCNets, which includes five components: (1) spatial graph convolutional layer, (2) feature transformation module, (3) supragraph convolutional module, (4) the DEPSRL module, and (5) GRU layer. The paper contributes non-trivial advancements over state-of-the-arts. The main concerns from the review’s end are as follows:
(1) How the learnable node embedding E_\phi is learned is not clear and the multiplication of X by E_\phi in Eqn. (1) is confusing.
(2) The mechanism to learn the normalized self-adaptive adjacency matrix S is similar to the self-attention mechanism used in GAT, which needs the input graph to be dense and which leads to inefficiency. In addition, \tilde{S} is a power series of S. The computation will be inefficiency. So the inefficiency is a big problem of this paper.
(3) In the ablation study, the paper does not compare using the normalized self-adaptive adjacency matrix S with using the pain adjacency matrix.


**Summary Of The Paper:**

The paper introduces multipersistence into graph neural networks (GNNs) to render GNNs capable of finding hidden time-conditioned patterns in spatio-temporal graph data. The proposed model, Time-Aware Multipersistence Spatio-Supra Graph Convolutional Network (TAMP-S2GCNets), outperforms other state-of-the-arts methods on various spatio-temporal graph data.

**Summary Of The Review:**

The paper’s contribution is non-trivial. Although the inefficiency may be a problem, I recommend accept.

---

### Official Review · Reviewer_T4LP · 2021-11-12

**Correctness:** 4
**Technical Novelty And Significance:** 4
**Empirical Novelty And Significance:** 4
**Recommendation:** 8
**Confidence:** 4

**Main Review:**

This paper proposes introduction of the multi-parameter persistence (as defined in the topological analysis)
to capture the time-dependent properties of spatio-temporal data and be a part of time-aware learning of
a convolutional network. The fusion of the two presents a novel research direction with improved performance over
state-of-the-art approaches.

Strong points:
- novel research direction (as explained above)
- the topic is of interest for the community. An effective approach of using multipersistence to allow
for capturing time dependencies and enhancing the learning of graphs.
- technically sound. The reviewer did their best to check the soundness of the technical backbone.
- Thorough experimental part. Code is provided and reproducability seems possible and well detailed in
the supplementary as well.
- The writing is clear. The motivation and is well stated and supported by also a very well presented
methodology.

Weak points:
Not many here.

- I have some questions regarding implementation choices that seem arbitrary- but probably
result of experimentation. In Eq(6) global average pooling and global max pooling are used. Some
details here would be nice.
- I would also add more analysis on the computation time compared to other methods. This is provided in
the supplementary material but I would consider moving the results in the main paper.




**Summary Of The Paper:**

The authors suggest the use of multiparameter persistence for
explicitly capturing he latent time dependencies in spatio-temporal data.
They propose the Time-Aware Multipersistence Spatio-Supra Graph
Convolutional Network that uses the mutlipersistence as extracted by the Euler-Poincare
surface and allows for time-aware learning.


**Summary Of The Review:**

A very interesting paper that proposes an approach that I expect to be impactful for future research.
I would like this paper to be part of the ICLR accepted works.

---

### Decision · Program_Chairs · 2022-01-20

**Decision:**

Accept (Spotlight)

**Comment:**

The authors introduce the Time-Aware Multiperistence Spatio-Supra Graph CN that uses
multiparameter persistence to capture the latent time dependencies in spatio-temporal data.

This is a novel and experimentally well-supported work. The novelty is achieved by combining research in topological analysis (multipersistence) and neural networks. Technically sound. Clear presentation and extensive experimental section.
Reviewers were uniformly positive, agreeing that the approach was interesting and well-motivated, and the experiments convincing. Some concerns that were raised were successfully addressed by the authors and revised in the manuscript.

Happy to recommend acceptance. A veru nice paper!